# CLASSIFIER-FREE GUIDANCE IS A PREDICTOR-CORRECTOR

## ABSTRACT

We investigate the theoretical foundations of classifier-free guidance (CFG). CFG is the dominant method of conditional sampling for text-to-image diffusion models, yet unlike other aspects of diffusion, it remains on shaky theoretical footing. In this paper, we first disprove common misconceptions, by showing that CFG interacts differently with DDPM (Ho et al., 2020) and DDIM (Song et al., 2021), and neither sampler with CFG generates the gamma-powered distribution $p(x|c)^\gamma p(x)^{1-\gamma}$. Then, we clarify the behavior of CFG by showing that it is a kind of predictor-corrector method (Song et al., 2020) that alternates between denoising and sharpening, which we call predictor-corrector guidance (PCG). We prove that in the SDE limit, CFG is actually equivalent to combining a DDIM predictor for the conditional distribution together with a Langevin dynamics corrector for a gamma-powered distribution (with a carefully chosen gamma). Our work thus provides a lens to theoretically understand CFG by embedding it in a broader design space of principled sampling methods.

## 1 INTRODUCTION

Classifier-free-guidance (CFG) has become an essential part of modern diffusion models, especially in text-to-image applications (Dieleman, 2022; Rombach et al., 2022; Nichol et al., 2021; Podell et al., 2023). CFG is intended to improve conditional sampling, e.g. generating images conditioned on a given class label or text prompt (Ho & Salimans, 2022). The traditional (non-CFG) way to do conditional sampling is to simply train a model for the conditional distribution $p(x \mid c)$, including the conditioning $c$ as auxiliary input to the model. In the context of diffusion, this means training a model to approximate the conditional score $s(x, t, c) := \nabla_x \log p_t(x \mid c)$ at every noise level $t$, and sampling from this model via a standard diffusion sampler (e.g. DDPM). Interestingly, this standard way of conditioning usually does not perform well for diffusion models, for reasons that are unclear. In the text-to-image case for example, the generated samples tend to be visually incoherent and not faithful to the prompt, even for large-scale models (Ho & Salimans, 2022; Rombach et al., 2022).

Guidance methods, such as CFG and its predecessor classifier guidance (Sohl-Dickstein et al., 2015; Song et al., 2020; Dhariwal & Nichol, 2021), are methods introduced to improve the quality of conditional samples. During training, CFG requires learning a model for both the unconditional and conditional scores ($\nabla_x \log p_t(x)$ and $\nabla_x \log p_t(x|c)$). Then, during sampling, CFG runs any standard diffusion sampler (like DDPM or DDIM), but replaces the true conditional scores with the "CFG scores"

$$\widetilde{s}(x, t, c) := \gamma \nabla_x \log p_t(x \mid c) + (1 - \gamma)\nabla \log p_t(x), \tag{1}$$

for some $\gamma > 0$. This turns out to produce much more coherent samples in practice, and so CFG is used in almost all modern text-to-image diffusion models (Dieleman, 2022). A common intuition for why CFG works starts by observing that Equation (1) is the score of a *gamma-powered* distribution:

$$p_{t,\gamma}(x|c) \propto p_t(x)^{1-\gamma} p_t(x|c)^\gamma, \tag{2}$$

which is also proportional to $p_t(x)p_t(c|x)^\gamma$. Raising $p_t(c|x)$ to a power $\gamma > 1$ sharpens the classifier around its modes, thereby emphasizing the "best" exemplars of the given class or other conditioner at each noise level. Applying CFG — that is, running a standard sampler with the usual score replaced by the CFG score at each denoising step — is supposed to increase the influence of the conditioner on the final samples.

CFG$_{\text{DDPM}}$ $\gamma = 2$ $\gamma = 5$ $\gamma = 10$ $\gamma = 20$

PCG$_{\text{DDIM}}$ $\gamma' = 3$ $\gamma' = 9$ $\gamma' = 19$ $\gamma' = 39$

Figure 1: **CFG vs. PCG**. We prove that the DDPM variant of classifier-free guidance (top) is equivalent to a kind of predictor-corrector method (bottom), in the continuous limit. We call this latter method "predictor-corrector guidance" (PCG), defined in Section 4.1. The equivalence holds for all CFG guidance strengths $\gamma$, with corresponding PCG parameter $\gamma' = (2\gamma - 1)$, as given in Theorem 3. Samples from SDXL with prompt: "photograph of a cat eating sushi using chopsticks".

However, CFG does not inherit the theoretical correctness guarantees of standard diffusion, because the CFG scores do not necessarily correspond to a valid diffusion forward process. The fundamental issue (which is known, but still worth emphasizing) is that $p_{t,\gamma}(x|c)$ is not the same as the distribution obtained by applying a forward diffusion process to the gamma-powered data distribution $p_{0,\gamma}(x|c)$. That is, letting $N_t[p]$ denote the distribution produced by starting from a distribution $p$ and running the diffusion forward process up to time $t$, we have

$$p_{t,\gamma}(x|c) := N_t[p_0(x|c)]^\gamma \cdot N_t[p_0(x)]^{1-\gamma} \neq N_t \left[ p_0(x|c)^\gamma p_0(x)^{1-\gamma} \right].$$

Since the distributions $\{p_{t,\gamma}(x|c)\}_t$ *do not correspond to any known forward diffusion process*, we cannot properly interpret the CFG score (1) as a denoising direction; and using the CFG score in a sampling loop like DDPM or DDIM is no longer theoretically guaranteed to produce a sample from $p_{0,\gamma}(x|c)$ or any other known distribution. Although this flaw is known in theory (e.g. Du et al. (2023); Karras et al. (2024a)), it is largely ignored in practice and in much of the literature. The theoretical foundations of CFG are thus unclear, and important questions remain open. Is there a principled way to think about why CFG works? And what does it even mean for CFG to "work" – what problem is CFG solving? We make progress towards understanding the foundations of CFG, and in the process we uncover several new aspects and connections to other methods.

1. First, we disprove common misconceptions about CFG by counterexample. We show that the DDPM and DDIM variants of CFG can generate different distributions, neither of which is the gamma-powered data distribution $p_0(x)^{1-\gamma} p_0(x|c)^\gamma$.

2. We define a family of methods called predictor-corrector guidance (PCG), as a natural way to approximately sample from gamma-powered distributions. PCG alternates between denoising steps and Langevin dynamics steps. In contrast to (Song et al., 2020), where the predictor and corrector both target the conditional distribution, in PCG the predictor anneals using conditional diffusion paths, while the corrector mixes towards the (sharpened) gamma-powered distribution.

3. We prove that in the continuous-time limit, CFG is equivalent to PCG with a careful choice of parameters. This gives a principled way to interpret CFG: it is implicitly an annealed Langevin dynamics.

4. For demonstration purposes, we implement the PCG sampler for Stable Diffusion XL and observe that it produces samples qualitatively similar to CFG, with guidance scales determined by our theory. Further, we explore the design axes exposed by the PCG framework, namely guidance strength and Langevin iterations, to clarify their respective effects.

## 2 PRELIMINARIES

We adopt the continuous-time stochastic differential equation (SDE) formalism of diffusion from Song et al. (2020). These continuous-time results can be translated to discrete-time algorithms; we give explicit algorithm descriptions for our experiments.

### 2.1 DIFFUSION SAMPLERS

Forward diffusion processes start with a conditional data distribution $p_0(x|c)$ and gradually corrupt it with Gaussian noise, with $p_t(x|c)$ denoting the noisy distribution at time $t$. The forward diffusion runs up to a time $T$ large enough that $p_T$ is approximately pure noise. To sample from the data distribution, we first sample from the Gaussian distribution $p_T$ and then run the diffusion process in reverse (which requires an estimate of the score, usually learned by a neural network). A variety of samplers have been developed to perform this reversal. DDPM (Ho et al., 2020) and DDIM (Song et al., 2021) are standard samplers that correspond to discretizations of a reverse-SDE and reverse-ODE, respectively. Due to this correspondence, we refer to the reverse-SDE as DDPM and the reverse-ODE as DDIM for short. The forward process, reverse-SDE, and equivalent reverse-ODE (Song et al., 2020) for the *variance-preserving* (VP) (Ho et al., 2020) conditional diffusion are

$$\text{Forward SDE}: dx = -\frac{1}{2}\beta_t x dt + \sqrt{\beta_t}dw. \tag{3}$$

$$\text{DDPM SDE}: \quad dx = -\frac{1}{2}\beta_t x\, dt - \beta_t \nabla_x \log p_t(x|c)dt + \sqrt{\beta_t}d\overline{w} \tag{4}$$

$$\text{DDIM ODE}: \quad dx = -\frac{1}{2}\beta_t x\, dt - \frac{1}{2}\beta_t \nabla_x \log p_t(x|c)dt. \tag{5}$$

The unconditional version of each sampler simply replaces $p_t(x|c)$ with $p_t(x)$. Note that the *score* $\nabla_x \log p_t(x|c)$ appears in both (4) and (5). Intuitively, the score points in a direction toward higher probability, and so it helps to reverse the forward diffusion process. The score is unknown in general, but can be learned via standard diffusion training methods.

### 2.2 CLASSIFIER-FREE GUIDANCE

CFG replaces the usual conditional score $\nabla_x \log p_t(x|c)$ in (4) or (5) at each timestep $t$ with the alternative score $\nabla_x \log p_{t,\gamma}(x|c)$. In SDE form, the CFG updates are

$$\text{CFG}_{\text{DDPM}}: \quad dx = -\frac{1}{2}\beta_t x\, dt - \beta_t \nabla_x \log p_{t,\gamma}(x|c)dt + \sqrt{\beta_t}d\overline{w} \tag{6}$$

$$\text{CFG}_{\text{DDIM}}: \quad dx = -\frac{1}{2}\beta_t x\, dt - \frac{1}{2}\beta_t \nabla \log p_{t,\gamma}(x|c)dt, \tag{7}$$

$$\text{where } \nabla_x \log p_{t,\gamma}(x|c) = (1-\gamma)\nabla_x \log p_t(x) + \gamma \nabla_x \log p_t(x|c).$$

### 2.3 LANGEVIN DYNAMICS

Langevin dynamics (Rossky et al., 1978; Parisi, 1981) is another sampling method, which starts from an arbitrary initial distribution and iteratively transforms it into a desired one. Langevin dynamics (LD) is given by the following SDE (Robert et al., 1999)

$$dx = \frac{\varepsilon}{2}\nabla \log \rho(x)dt + \sqrt{\varepsilon}dw. \tag{8}$$

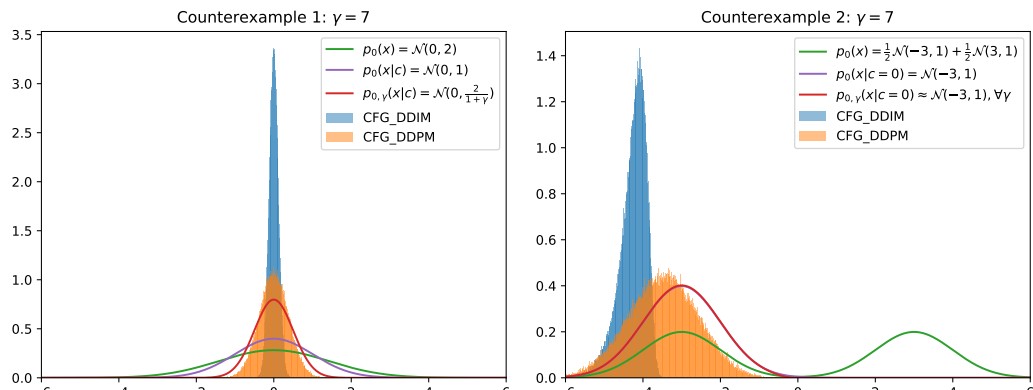

Figure 2: **Counterexamples: CFG$_{\text{DDIM}} \neq$ CFG$_{\text{DDPM}} \neq$ gamma-powered.** CFG$_{\text{DDIM}}$ and CFG$_{\text{DDPM}}$ do not generate the same output distribution, even when using the same score function. Moreover, neither generated distribution is the gamma-powered distribution $p_{0,\gamma}(x|c)$. (Left) Counterexample 1 (section 3.1): CFG$_{\text{DDIM}}$ yields a sharper distribution than CFG$_{\text{DDPM}}$, and both are sharper than $p_{0,\gamma}(x|c)$. (Right) Counterexample 2 (section 3.2): Neither CFG$_{\text{DDIM}}$ nor CFG$_{\text{DDPM}}$ yield even a scaled version of the gamma-powered distribution $p_{0,\gamma}(x|c) = \mathcal{N}(-3, 1)$. The CFG$_{\text{DDPM}}$ distribution is mean-shifted relative to $p_{0,\gamma}(x|c)$. The CFG$_{\text{DDIM}}$ distribution is mean-shifted and not even Gaussian (note the asymmetrical shape).

LD converges (under some assumptions) to the steady-state $\rho(x)$ (Roberts & Tweedie, 1996). That is, letting $\rho_s(x)$ denote the solution of LD at time $s$, we have $\lim_{s \to \infty} \rho_s(x) = \rho(x)$. Similar to diffusion sampling, LD requires the score of the desired distribution $\rho$ (or a learned estimate of it).

## 3 Misconceptions about CFG

We first observe that the exact definition of CFG matters: specifically, the sampler with which it used. Without CFG, DDPM and DDIM generate equivalent distributions. However, we will prove that with CFG, DDPM and DDIM can generate different distributions. We provide informal statements of our claims below, to convey the main intuitions. The formal statement and proof is provided in Appendix A.1, as Theorem 4.

**Theorem 1** (CFG$_{\text{DDIM}} \neq$ CFG$_{\text{DDPM}}$; informal). *Consider generating a sample via CFG using either DDPM or DDIM as the sampler. There exists a particular data distribution for which the generations of CFG differ depending on the choice of sampler. In particular, for large guidance scale $\gamma \gg 1$, CFG$_{\text{DDPM}}$ and CFG$_{\text{DDPM}}$ approximately generate the following distributions, respectively:*

$$\widehat{p}_{\text{ddpm}} \approx \mathcal{N}(0, \gamma^{-1}), \quad \widehat{p}_{\text{ddim}} \approx \mathcal{N}(0, 2^{-\gamma}).$$

Next, we disprove the misconception that CFG generates the gamma-powered distribution data:

**Theorem 2** (CFG $\neq$ gamma-sharpening, informal). *There exists a data distribution $p_0$ such that for any $\gamma > 0$, neither CFG$_{\text{DDIM}}$ nor CFG$_{\text{DDPM}}$ produces the gamma-powered distribution $p_{0,\gamma}(x|c) \propto p_0(x)^{1-\gamma} p_0(x|c)^{\gamma}$.*

Both claims are proven using a simple Gaussian construction, as outlined in the next section.

### 3.1 Counterexample 1

We first present a setting that allows us to *exactly* solve the ODE and SDE dynamics of CFG in closed-form, and hence to find the exact distribution sampled by running CFG. This would be intractable in general, but it is possible for a specific problem, as follows.

Consider a setting where $p_0(x)$ and $p_0(x|c = 0)$ are both zero-mean Gaussians, but with different variances. Specifically, $(x_0, c)$ are jointly Gaussian, with $p(c) = \mathcal{N}(0, 1)$, $p_0(x|c) = c + \mathcal{N}(0, 1)$.

Therefore

$$p_0(x) = \mathcal{N}(0, 2)$$
$$p_0(x|c = 0) = \mathcal{N}(0, 1)$$
$$p_{0,\gamma}(x|c = 0) = \mathcal{N}(0, \frac{2}{\gamma + 1}) \tag{9}$$

For this problem, we can solve $\mathsf{CFG_{DDIM}}$ (7) and $\mathsf{CFG_{DDPM}}$ (6) analytically; that is, we solve initial-value problems for the reversed dynamics to find the sampled distribution of $\widehat{x}_t$ in terms of the initial-value $x_T$. Applying these results to $t = 0$ and averaging over the known Gaussian distribution of $x_T$ gives the exact distribution of $\widehat{x}_0$ that CFG samples. The full derivation is in Appendix A.1. The final CFG-sampled distributions are:

$$\mathsf{CFG_{DDPM}}: \quad \widehat{x}_0 \sim \mathcal{N}\left(0, \frac{2 - 2^{2-2\gamma}}{2\gamma - 1}\right) \tag{10}$$

$$\mathsf{CFG_{DDIM}}: \quad \widehat{x}_0 \sim \mathcal{N}\left(0, 2^{1-\gamma}\right). \tag{11}$$

This shows that for any $\gamma > 1$, the $\mathsf{CFG_{DDIM}}$ distribution is sharper than the $\mathsf{CFG_{DDPM}}$ distribution, and both are sharper than the gamma-powered distribution $p_{0,\gamma}(x|c = 0)$. (Even though the distributions all have the same mean, their different variances make them distinct.) In fact, for $\gamma \gg 1$, the variance of DDPM-CFG is approximately $\frac{2}{2\gamma - 1}$, which is about twice the variance of $p_{0,\gamma}(x|c = 0)$. In Figure 2, we compare the $\mathsf{CFG_{DDIM}}$ and $\mathsf{CFG_{DDPM}}$ distributions – sampled using an exact denoiser (see Appendix A.6) within DDIM/DDPM sampling loops – to the unconditional, conditional, and gamma-powered distributions.

### 3.2 COUNTEREXAMPLE 2

In the above counterexample, the $\mathsf{CFG_{DDIM}}$, $\mathsf{CFG_{DDPM}}$, and gamma-powered distributions had different variances but the same Gaussian form, so one might wonder whether the distributions differ only by a scale factor in general. This is not the case, as we can see in a different counterexample that reveals greater qualitative differences, in particular a symmetry-breaking behavior of CFG.

In Counterexample 2, the unconditional distribution is a Gaussian mixture with two clusters with equal weights and variances, and means at $\pm\mu$.

$$c \in \{0, 1\}, \quad p(c = 0) = \frac{1}{2}$$
$$p_0(x_0|c = 0) = \mathcal{N}(-\mu, 1), \quad p_0(x_0|c = 1) = \mathcal{N}(\mu, 1)$$
$$p_0(x_0) = \frac{1}{2}p_0(x_0|c = 0) + \frac{1}{2}p_0(x_0|c = 1) \tag{12}$$

If the means are sufficiently separated ($\mu \gg 1$), then the gamma-powered distribution for $\gamma \geq 1$ is approximately equal to the conditional distribution, i.e. $p_{0,\gamma}(x|c) \approx p_0(x|c)$, due to the near-zero-probability valley between the conditional densities (see Appendix A.2). However, for sufficiently high noise the clusters begin to merge, and $p_{t,\gamma}(x|c) \neq p_t(x|c)$. In particular, $p_{0,\gamma}(x|c)$ is approximately Gaussian with mean $\pm\mu$, but $p_{t,\gamma}(x|c) \neq p_t(x|c)$ is not. Although we cannot solve the CFG ODE and SDE in this case, we can empirically sample the $\mathsf{CFG_{DDIM}}$ and $\mathsf{CFG_{DDPM}}$ distributions using an exact denoiser and compare them to the gamma-powered distribution. In particular, we see that neither $\mathsf{CFG_{DDIM}}$ nor $\mathsf{CFG_{DDPM}}$ is Gaussian with mean $\pm\mu$, hence neither is a scaled version of the gamma-powered distribution. The results are shown in Figure 2. Concurrent work by Chidambaram et al. (2024) offers a theoretical analysis confirming our qualitative observations in the two-cluster case, while Wu et al. (2024) conduct an analysis of similar GMM settings.

## 4 CFG AS A PREDICTOR-CORRECTOR

The previous sections illustrated the subtlety in understanding CFG. We can now state our main structural characterization, that CFG is equivalent to a special kind of *predictor-corrector* method (Song et al., 2020).

## 4.1 PREDICTOR-CORRECTOR GUIDANCE

As a warm-up, suppose we actually wanted to sample from the gamma-powered distribution:

$$p_\gamma(x|c) \propto p(x)^{1-\gamma} p(x|c)^\gamma. \tag{13}$$

A natural strategy is to run Langevin dynamics w.r.t. $p_\gamma$. This is possible in theory because we can compute the score of $p_\gamma$ from the known scores of $p(x)$ and $p(x \mid c)$:

$$\nabla_x \log p_\gamma(x \mid c) = (1 - \gamma)\nabla_x \log p(x) + \gamma \nabla_x \log p(x \mid c). \tag{14}$$

However this won't work in practice, due to the well-known issue that vanilla Langevin dynamics has impractically slow mixing times for many distributions of interest (Song & Ermon, 2019). The usual remedy for this is to use some kind of annealing, and the success of diffusion teaches us that the diffusion process defines a good annealing path (Song et al., 2020; Du et al., 2023). Combining these ideas yields an algorithm remarkably similar to the predictor-corrector methods introduced in Song et al. (2020). For example, consider the following diffusion-like iteration, starting from $x_T \sim \mathcal{N}(0, \sigma_T)$ at $t = T$. At timestep $t$,

1. Predictor: Take one diffusion denoising step (e.g. DDIM or DDPM) w.r.t. $p_t(x \mid c)$, using score $\nabla_x \log p_t(x \mid c)$, to move to time $t' = t - \Delta t$.

2. Corrector: Take $K \geq 1$ Langevin dynamics steps w.r.t. distribution $p_{t',\gamma}$, using score

$$\nabla_x \log p_{t',\gamma}(x \mid c) = (1 - \gamma)\nabla_x \log p_{t'}(x) + \gamma \nabla_x \log p_{t'}(x \mid c).$$

It is reasonable to expect that running this iteration down to $t = 0$ will produce a sample from approximately $p_\gamma(x|c)$, since the iteration can be thought of as a type of annealed Langevin dynamics, with time $t$ playing the role of temperature (c.f. Remark 1 below). We name this algorithm predictor-corrector guidance (PCG). Remarkably, it turns out that for specific choices of the denoising predictor and Langevin step size, PCG is equivalent (in the SDE limit) to CFG, but with a different $\gamma$. We will formalize and prove this in the subsequent section.

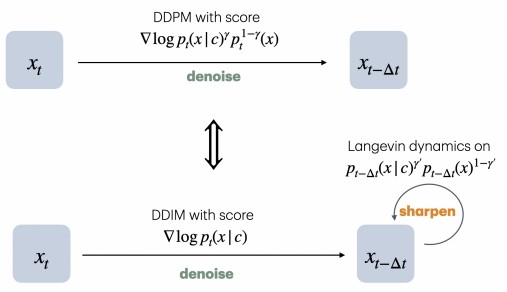

Figure 3: CFG is equivalent to PCG for particular parameter choices.

**Remark 1** (Langevin Dynamics). *The standard annealed Langevin dynamics corresponds to a predictor-corrector where the predictor is an identity function: it only reduces the "temperature" $t \to t - \Delta t$ without changing the current sample $x_t$. The iteration above uses an intuitively better predictor that moves $x_t$ along the diffusion path, which is the "correct" way to reduce temperature (at least in the conditional diffusion setting).*

**Remark 2** (Mixing). *Why do we expect PCG to sample from approximately $p_\gamma(x|c)$? For the same reason we expect annealed Langevin dynamics to work: in the limit of many Langevin steps ($K \to \infty$), the corrector will fully mix to the distribution $p_{t',\gamma}$ at each time $t'$. In reality we may take only $K = 1$ Langevin step at each iteration, which will at least move the sample distribution towards the target distribution $p_{t',\gamma}(x|c)$, even if it does not fully mix.*

**Remark 3** (Predictor-Corrector). *PCG technically differs from the predictor-corrector algorithms in Song et al. (2020), because our predictor and corrector operate w.r.t. different distributions ($p_t$ vs. $p_{t,\gamma}$). However, conceptually all of these methods can be thought of as variants of annealed Langevin dynamics (as described in Remark 1), with different annealing choices.*

## 4.2 SDE LIMIT OF PCG

Consider the version of PCG defined in Algorithm 1, which uses DDIM as predictor and a particular LD on the gamma-powered distribution as corrector. We take $K = 1$, i.e. a single LD step per iteration. Crucially, we set the LD step size such that the Langevin noise scale exactly matches the

---

**Algorithm 1:** PCG$_\text{DDIM}$, theory. (see Algorithm 2 for practical implementation.)

---

**Input:** Conditioning $c$, guidance weight $\gamma \geq 0$
**Constants:** $\beta_t := \beta(t)$ from Song et al. (2020). $K \in \mathbb{N}$, the number of Langevin iterations.

1   $x_1 \sim \mathcal{N}(0, I)$
2   **for** $(t = 1 - \Delta t;\ t \geq 0;\ t \leftarrow t - \Delta t)$ **do**
3     $s_{t+\Delta t} := \nabla \log p_{t+\Delta t}(x_{t+\Delta t}|c)$
4     $x_t \leftarrow x_{t+\Delta t} + \frac{1}{2}\beta_t(x_{t+\Delta t} + s_{t+\Delta t})\Delta t$         $\triangleright$ DDIM step for $p_{t+\Delta t}(x|c) \to p_t(x|c)$
5     $\varepsilon := \beta_t \Delta t$           $\triangleright$ Langevin step size, matching DDPM noise scale $\beta_t$
6     **for** $k = 1, \ldots K$ **do**
7       $\eta \sim \mathcal{N}(0, I_d)$
8       $s_{t,\gamma} := (1 - \gamma)\nabla \log p_t(x_t) + \gamma \nabla \log p_t(x_t|c)$
9       $x_t \leftarrow x_t + \frac{\varepsilon}{2}s_{t,\gamma} + \sqrt{\varepsilon}\eta$          $\triangleright$ Langevin dynamics on $p_{t,\gamma}(x|c)$
10    **end**
11 **end**
12 **return** $x_0$

---

noise scale of a (hypothetical) DDPM step at the current time (similar to Du et al. (2023)). In the limit as $\Delta t \to 0$, Algorithm 1 becomes the following SDE (see Appendix B):

$$dx = \underbrace{\Delta\text{DDIM}(x,t)}_{\text{Predictor}} + \underbrace{\Delta\text{LD}_\text{G}(x,t,\gamma)}_{\text{Corrector}} =: \Delta\text{PCG}_\text{DDIM}(x,t,\gamma), \tag{15}$$

where $\Delta\text{DDIM}(x,t) = -\frac{1}{2}\beta_t(x + \nabla \log p_t(x|c))dt$

$$\Delta\text{LD}_\text{G}(x,t,\gamma) = -\frac{1}{2}\beta_t\Big((1-\gamma)\nabla \log p_t(x) + \gamma\nabla \log p_t(x|c)\Big)dt + \sqrt{\beta_t}d\overline{w}.$$

Above, $\Delta\text{DDIM}(x,t)$ is the *differential* of the DDIM ODE (5), i.e. the ODE can be written as $dx = \Delta\text{DDIM}(x,t)$. And $\Delta\text{LD}_\text{G}(x,t,\gamma)$, where G stands for "guidance", is the limit as $\Delta t \to 0$ of the Langevin dynamics step in PCG, which behaves like a differential of LD (see Appendix B).

We can now show that the PCG SDE (15) matches CFG with DDPM, but with a different $\gamma$. In the statement, $\Delta\text{CFG}_\text{DDPM}(x,t,\gamma)$ denotes the differential of the CFG$_\text{DDPM}$ SDE (6), similar to the notation above. This result is trivial to prove using our definitions, but the statement itself appears to be novel. [1]

**Theorem 3** (CFG is predictor-corrector). *In the SDE limit, CFG with DDPM is equivalent to a predictor-corrector. That is, the following differentials are equal:*

$$\Delta\text{CFG}_\text{DDPM}(x,t,\gamma) = \Delta\text{DDIM}(x,t) + \Delta\text{LD}_\text{G}(x,t,2\gamma - 1) =: \Delta\text{PCG}_\text{DDIM}(x,t,2\gamma - 1) \tag{16}$$

*Notably, the guidance scales of CFG and the above Langevin dynamics are not identical.*

*Proof.*

$$\Delta\text{PCG}_\text{DDIM}(x,t,\gamma) = \Delta\text{DDIM}(x,t) + \Delta\text{LD}_\text{G}(x,t,\gamma)$$
$$= -\frac{1}{2}\beta_t(x + (1-\gamma)\nabla \log p_t(x) + (1+\gamma)\nabla \log p_t(x|c))dt + \sqrt{\beta_t}d\overline{w}$$
$$= -\frac{1}{2}\beta_t x \Delta t - \beta_t \nabla_x \log p_{t,\gamma'}(x|c)\Delta t + \sqrt{\beta_t}d\overline{w}, \quad \gamma' := \frac{\gamma}{2} + \frac{1}{2}$$
$$= \Delta\text{CFG}_\text{DDPM}(x,t,\gamma')$$

$\square$

---

[1] Notice that taking $\gamma = 1$ in Theorem 3 recovers the standard fact that DDPM is equivalent, in the limit, to DDIM interleaved with LD (e.g. Karras et al. (2022)). This is because for $\gamma = 1$, CFG$_\text{DDPM}$ is just DDPM, so Theorem 3 reduces to: $\Delta\text{DDPM}(x,t) = \Delta\text{DDIM}(x,t) + \Delta\text{LD}_\text{G}(x,t,1)$.

## 5 DISCUSSION AND RELATED WORKS

There have been many recent works toward understanding CFG. To better situate our work, it helps to first discuss the overall research agenda.

### 5.1 UNDERSTANDING CFG: THE BIG PICTURE

We want to study the question of why CFG helps in practice: specifically, why it improves both image quality and prompt adherence, compared to conditional sampling. We can approach this question by applying a standard generalization decomposition. Let $p(x|c)$ be the "ground truth" population distribution; let $p^*_\gamma(x|c)$ be the distribution generated by the ideal CFG sampler, which exactly solves the CFG reverse SDE for the ground-truth scores (note that at $\gamma = 1$, $p^*_1(x|c) = p(x|c)$); and let $\widehat{p}_\gamma(x|c)$ denote the distribution of the real CFG sampler, with learnt scores and finite discretization. Now, for any image distribution $q$, let $\mathsf{PerceivedQuality}[q] \in \mathbb{R}$ denote a measure of perceived sample quality of this distribution to humans. We cannot mathematically specify this notion of quality, but we will assume it exists for analysis. Notably, $\mathsf{PerceivedQuality}$ is *not* a measurement of how close a distribution is to the ground-truth $p(x|c)$ — it is possible for a generated distribution to appear even "higher quality" than the ground-truth, for example. We can now decompose:

$$\underbrace{\mathsf{PerceivedQuality}[\widehat{p}_\gamma]}_{\text{Real CFG}} = \underbrace{\mathsf{PerceivedQuality}[p^*_\gamma]}_{\text{Ideal CFG}} - \underbrace{\left(\mathsf{PerceivedQuality}[p^*_\gamma] - \mathsf{PerceivedQuality}[\widehat{p}_\gamma]\right)}_{\text{Generalization Gap}} . \tag{17}$$

Therefore, if the LHS increases with $\gamma$, it must be because at least one of the two occurs:

1. The ideal CFG sampler improves in quality with increasing $\gamma$. That is, CFG distorts the population distribution in a favorable way (e.g. by sharpening it, or otherwise).

2. The generalization gap decreases with increasing $\gamma$. That is, CFG has a type of regularization effect, bringing population and empirical processes closer.

In fact, it is likely that both occur. The original motivation for CG and CFG involved the first effect: CFG was intended to produce "lower-temperature" samples from a sharpened population distribution (Dhariwal & Nichol, 2021; Ho & Salimans, 2022). This is particularly relevant if the model is trained on poor-quality datasets (e.g. cluttered images from the web), so we want to use guidance to sample from a higher-quality distribution (e.g. images of an isolated subject). On the other hand, recent studies have given evidence for the second effect. For example, Karras et al. (2024a) argues that unguided diffusion sampling produces "outliers," which are avoided when using guidance — this can be thought of as guidance reducing the generalization gap, rather than improving the ideal sampling distribution. Another interpretation of the second effect is that guidance could enforce a good inductive bias: it "simplifies" the family of possible output distributions in some sense, and thus simplifies the learning problem, reducing the generalization gap. Figure 6 shows a example where this occurs. Finally, this generalization decomposition applies to any intervention to the SDE, not just increasing guidance strength. For example, increasing the Langevin steps in PCG (parameter $K$) also shrinks the generalization gap, since it reduces the discretization error.

In this framework, our work makes progress towards understanding both terms on the RHS of Equation 17, in different ways. For the first term, we identify structural properties of ideal CFG, by showing that $p^*_\gamma$ can be equivalently generated by a standard technique (an annealed Langevin dynamics). For the second term, the PCG framework highlights the ways in which errors in the learned score can contribute to a generalization gap, during both the denoising step and the LD step (the latter would move toward an inaccurate steady-state distribution).

### 5.2 OPEN QUESTIONS AND LIMITATIONS

In addition to the above, there are a number of other questions left open by our work. First, we study only the stochastic variant of CFG (i.e. $\mathsf{CFG_{DDPM}}$), and it is not clear how to adapt our analysis to the more commonly used deterministic variant ($\mathsf{CFG_{DDIM}}$). This is subtle because the two CFG

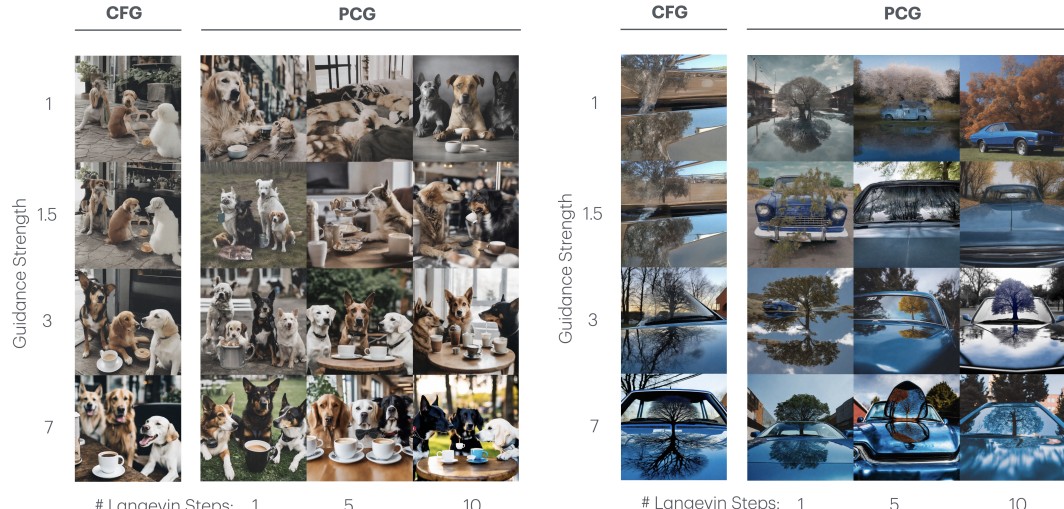

Figure 4: **Effect of Guidance and Correction.** Each grid shows SDXL samples using $PCG_{DDIM}$, as the guidance strength $\gamma$ and Langevin iterations $K$ are varied. Left: "photograph of a dog drinking coffee with his friends". Right: "a tree reflected in the hood of a blue car". (Zoom in to view).

variants can behave very differently in theory, but appear to behave similarly in practice. It is thus open to identify plausible theoretical conditions which explain this similarity[2]; we give a suggestive experiment in Figure 5. More broadly, it is open to find more explicit characterizations of CFG's output distribution, in terms of the original $p(x)$ and $p(x|c)$.

Finally, we presented PCG primarily as a tool to understand CFG, not as a practical algorithm in itself. Nevertheless, the PCG framework outlines a broad family of guided samplers, which may be promising to explore in practice. For example, the predictor can be any diffusion denoiser, including CFG itself. The corrector can operate on any distribution with a known score, including compositional distributions as in Du et al. (2023), or any other distribution that might help sharpen or otherwise improve on the conditional distribution. Finally, the number of Langevin steps could be adapted to the timestep, similar to Kynkäänniemi et al. (2024), or alternative samplers could be considered (Du et al., 2023; Neal, 2012; Ma et al., 2015).

### 5.3 STABLE DIFFUSION EXAMPLES

We include several examples running predictor-corrector guidance on Stable Diffusion XL (Podell et al., 2023). These serve primarily to sanity-check our theory, not as a suggestion for practice. For all experiments, we use $PCG_{DDIM}$ as implemented explicitly in Algorithm 2. Note that PCG offers a more flexible design space than standard CFG; e.g. we can run multiple corrector steps for each denoising step to improve the quality of samples (controlled by parameter $K$ in Algorithm 2).

**CFG vs. PCG.** Figure 1 illustrates the equivalence of Theorem 3: we compare $CFG_{DDPM}$ with guidance $\gamma$ to $PCG_{DDIM}$ with exponent $\gamma' := (2\gamma - 1)$. We run $CFG_{DDPM}$ with 200 denoising steps, and $PCG_{DDIM}$ with 100 denoising steps and $K = 1$ Langevin step per denoising step. Corresponding samples appear to have qualitatively similar guidance strengths, consistent with our theory.

**Effects of Guidance and Corrector.** In Figure 4 we show samples from $PCG_{DDIM}$, varying the guidance strength and Langevin iterations (i.e. parameters $\gamma$ and $K$ respectively in Algorithm 2). We also include standard $CFG_{DDIM}$ samples for comparison. All samples used 1000 denoising steps for the base predictor. Overall, we observed that increasing Langevin steps tends to improve the overall image quality, while increasing guidance strength tends to improve prompt adherence. In particular, sufficiently many Langevin steps can sometimes yield high-quality conditional samples, even *without*

---

[2]Curiously, $CFG_{DDIM}$ is the correct probability-flow ODE for $CFG_{DDPM}$ if and only if the true intermediate distribution at time $t$ is $p_{t,\gamma}$. However we know this is not the true distribution in general, from Section 3.

| Method | $\gamma = 1$ | $\gamma = 1.1$ | $\gamma = 1.3$ | $\gamma = 1.5$ |
|---|---|---|---|---|
| CFG$_{\text{DDPM}}$ | 5.99 | 3.90 | 2.71 | 3.33 |
| CFG$_{\text{DDIM}}$ | 7.11 | 4.61 | 2.55 | 2.47 |
| PCG$_{\text{DDIM}}$ (LD steps=1) | 7.77 | 5.54 | 3.37 | 3.16 |
| PCG$_{\text{DDIM}}$ (LD steps=3) | 7.42 | 4.11 | 3.71 | 6.10 |
| PCG$_{\text{DDIM}}$ (LD steps=5) | 7.23 | 3.80 | 4.87 | 8.86 |

Table 1: FID Scores on ImageNet (lower is better), using DDPM, DDIM, and PCG samplers. We vary $\gamma$ and the number of LD steps. FD-DINOv2 and Inception Scores provided in Appendix C.

any guidance ($\gamma = 1$); see Figure 7 in the Appendix for another such example. This is consistent with the observations of Song et al. (2020) on unguided predictor-corrector methods. It is also related to the findings of Du et al. (2023) on MCMC methods: Du et al. (2023) similarly use an annealed Langevin dynamics with reverse-diffusion annealing, although they focus on general compositions of distributions rather than the specific gamma-powered distribution of CFG.

Notice that in Figure 4, increasing the number of Langevin steps appears to also increase the "effective" guidance strength. This is because the dynamics does not fully mix: one Langevin step ($K = 1$) does not suffice to fully converge the intermediate distributions to $p_{t,\gamma}$.

### 5.4 IMAGENET EXPERIMENTS

For completeness, we also include experiments comparing variants of PCG and CFG on ImageNet (Russakovsky et al., 2015). Table 1 shows FID scores (Heusel et al., 2017) on ImageNet, using EDM2 pretrained diffusion models (Karras et al., 2024b). Metrics are calculated using 50,000 samples and 200 sampling steps, generated using EDM2 checkpoints `edm2-img512-s-2147483-0.025` (conditional) and `edm2-img512-xs-uncond-2147483-0.025` (unconditional).

- For all samplers, there is a "sweet spot" of guidance scale $\gamma$; quality starts to degrade when $\gamma$ is too low or too high. This is a well-known behavior of CFG, and also occurs for PCG.
- For PCG methods, increasing the number of LD steps does not always improve FID — it depends on the guidance scale. More LD steps helps at $\gamma = 1.1$ for example, but starts to hurt at higher $\gamma$. This may seem surprising, but is explained by the same mechanism we saw in Figure 4: increasing the LD steps corresponds to increasing the "effective" guidance strength, because a single step does not fully mix the Langevin dynamics.
- CFG$_{\text{DDPM}}$ and PCG$_{\text{DDIM}}$ (LD=1) have different optimal guidance scales $\gamma$. The FID of CFG$_{\text{DDPM}}$ is minimized at $\gamma \approx 1.3$, while PCG$_{\text{DDIM}}$ is minimized at $\geq 1.5$. This is roughly in line with Theorem 3, where the equivalence between PCG and CFG requires rescaling $\gamma$.
- Finally, for $\gamma = 1$, both PCG$_{\text{DDIM}}$ and CFG$_{\text{DDPM}}$ are equivalent to standard DDPM in the SDE limit. However, PCG$_{\text{DDIM}}$ has significantly worse FID in the above finite-stepsize experiment. This discrepancy can thus be attributed to different discretization strategies of the same SDE — similar to how DDPM is a more sophisticated discretization than Euler–Maruyama for the reverse-diffusion SDE (e.g. Lu et al. (2022b)).

## 6 CONCLUSION

We have shown that while CFG is not a diffusion sampler on the gamma-powered data distribution $p_0(x)^{1-\gamma} p_0(x|c)^{\gamma}$, it can be understood as a particular kind of predictor-corrector, where the predictor is a DDIM denoiser, and the corrector at each step $t$ is one step of Langevin dynamics on the gamma-powered noisy distribution $p_t(x)^{1-\gamma'} p_t(x|c)^{\gamma'}$, with $\gamma' = (2\gamma - 1)$. Although Song et al. (2020)'s Predictor-Corrector algorithm has not been widely adopted in practice, perhaps due to its computation expense relative to samplers like DPM++ (Lu et al., 2022b), it turns out to provide a lens to understand the unreasonable practical success of CFG. On a practical note, PCG encompasses a rich design space of possible predictors and correctors for future exploration, that may help improve the prompt-alignment, diversity, and quality of diffusion generation.

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

## A   1D GAUSSIAN COUNTEREXAMPLES

In this section, we formalize and prove Theorems 1 and 2. We will work with a *variance-exploding* (VE) process, so we begin by defining CFG for the VE process (analogous to the SDE (6) and ODE (7) for the VP process).

**Definition 1** (CFG, variance-exploding). *Given a data distribution $p_0(x, c)$, define the noisy distribution $p_t(x)$ for any $t \in \mathbb{R}_+$ as the result of running the VE forward diffusion SDE $dx = dw$, up to time $t$, with initial distribution $p_0(x)$ at $t = 0$. Explicitly, this is the convolution $p_t := p_0 \star \mathcal{N}(0, t)$. Similarly define $p_t(x|c) := p_0(x|c) \star \mathcal{N}(0, t)$.*

*For all $\gamma \in \mathbb{R}$ and $c \in \mathbb{R}$, define the CFG SDEs for DDPM and DDIM, respectively, as*

$$CFG_{DDPM} : \quad dx = -\nabla_x \log p_{t,\gamma}(x|c)dt + d\overline{w}, \tag{18}$$

$$CFG_{DDIM} : \quad \frac{dx}{dt} = -\frac{1}{2}\nabla_x \log p_{t,\gamma}(x|c), \tag{19}$$

*where $p_{t,\gamma}(x|c) := p_t(x|c)^\gamma p_t(x)^{1-\gamma}/Z$, and $Z \in \mathbb{R}_+$ is the appropriate normalization constant.*

The SDE and ODE above specify the dynamics of the CFG sampler in the VE setting. Specifically, in order to sample via CFG, we start with a Gaussian sample $x_T \sim \mathcal{N}(0, T)$ for some $T \gg 0$, and then run the SDE or ODE from time $t = T$ down to time $t = 0$, to generate a sample $x_0$. We call the resulting distribution of samples $x_0$ the *generated distribution*, and adopt the following notation:

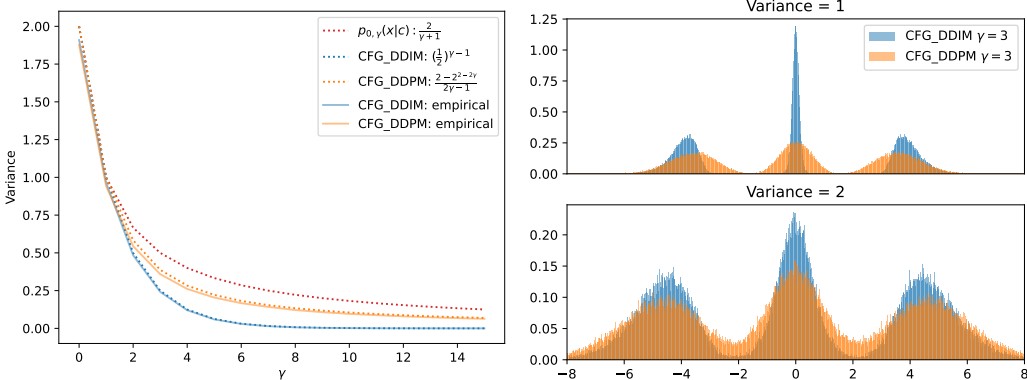

Figure 5: (Left) For Counterexample 1 (section 3.1), we plot the empirical and theoretical variance of the gamma-powered, $\mathsf{CFG_{DDIM}}$, and $\mathsf{CFG_{DDPM}}$ distributions, over a range of values of $\gamma$. The theoretical predictions are given by equations (11) and (10), and the empirical distributions are sampled using an exact denoiser. This verifies the theoretical predictions and illustrates the decreasing variance from $p_{0,\gamma}$ to $\mathsf{CFG_{DDPM}}$ to $\mathsf{CFG_{DDIM}}$. (Right) For counterexample 3 (section A.3 with different choices of variance ($\sigma = 1$ and $\sigma = 2$), we compare $\mathsf{CFG_{DDIM}}$ and $\mathsf{CFG_{DDPM}}$. Increasing the variance makes the two CFG samplers more similar. Also note that the $\mathsf{CFG_{DDIM}}$ distribution is symmetric around the center cluster, but asymmetric around the side clusters. This experiment suggests that multiple clusters and greater overlap between classes can help symmetrize and reduce the difference between $\mathsf{CFG_{DDIM}}$ and $\mathsf{CFG_{DDPM}}$

**Definition 2** (CFG generated distributions). *Denote by $p_{DDPM}^{(T)}(x|c)$, $p_{DDIM}^{(T)}(x|c)$ the probability densities of the distributions generated by the $\mathsf{CFG_{DDPM}}$ SDE (18), $\mathsf{CFG_{DDIM}}$ ODE (19), respectively; that is, the solutions to the SDE, ODE, respectively, at time $t = 0$ with initial conditions $x_T \sim \mathcal{N}(0, T)$, for any terminal times $T \in \mathbb{R}_+$ and conditioning $c \in \mathbb{R}$.*

We will mainly be interested in the limits of the generated distributions as we let the terminal time $T \to \infty$, which corresponds to allowing the diffusion process to fully mix. We can now formalize Theorems 1 and 2 as follows:

**Theorem 4** (Counterexample for which $\mathsf{CFG_{DDIM}} \neq \mathsf{CFG_{DDPM}} \neq$ gamma-sharpening). *In the setting of Definitions 1 and 2, there exists a data distribution such that the distributions generated by $\mathsf{CFG_{DDPM}}$ and $\mathsf{CFG_{DDIM}}$ are different, and neither is equal to the gamma-powered distribution. Specifically, define a data distribution $p_0(x, c)$, over inputs $x \in \mathbb{R}$ and conditioning $c \in \mathbb{R}$, as:*

$$p_0(c) = \mathcal{N}(c; 0, 1), \quad p_0(x|c) = \mathcal{N}(x; c, 1).$$

*In particular, $(x, c) \in \mathbb{R}^2$ is jointly Gaussian and $p_0(x|c = 0) = \mathcal{N}(x; 0, 1)$.*

*Then, for all $x, \gamma \in \mathbb{R}$, the limiting generated distributions for $c = 0$ are:*

$$\lim_{T \to \infty} p_{DDPM}^{(T)}(x|c = 0) = \mathcal{N}\left(x; 0, \frac{2 - 2^{2-2\gamma}}{2\gamma - 1}\right) \tag{20}$$

$$\lim_{T \to \infty} p_{DDIM}^{(T)}(x|c = 0) = \mathcal{N}\left(x; 0, 2^{1-\gamma}\right). \tag{21}$$

*Furthermore, the gamma-powered distribution for $c = 0$ is given by $p_{0,\gamma}(x|c = 0) = \mathcal{N}(x; 0, \frac{2}{\gamma+1})$. Therefore,*

$$\lim_{T \to \infty} p_{DDPM}^{(T)}(x|c = 0) \neq \lim_{T \to \infty} p_{DDIM}^{(T)}(x|c = 0) \neq p_{0,\gamma}(x|c = 0).$$

Note that variance of the generated distributions depends on the guidance weight $\gamma$ (Equations 20 and 20), and is exponentially different between DDIM and DDPM when $\gamma \gg 1$. The proof follows directly from the calculations in the next section (A.1), which characterize the density evolution of CFG in this setting.

## A.1  COUNTEREXAMPLE 1

Counterexample 1 (equation 9) has

$$p(c) = \mathcal{N}(0,1)$$
$$p_0(x|c) = \mathcal{N}(c,1)$$
$$\implies p_0(x) \sim \mathcal{N}(0,2)$$
$$p_0(x|c=0) \sim \mathcal{N}(0,1).$$

The $\gamma$-powered distribution is

$$p_{0,\gamma}(x|c=0) = p_0(x|c)^{\gamma} p_{c=0}(x)^{1-\gamma}$$
$$\propto e^{-\frac{\gamma x^2}{2}} e^{-\frac{(1-\gamma)x^2}{4}} = e^{-\frac{(\gamma+1)x^2}{4}}$$
$$\sim \mathcal{N}(0, \frac{2}{\gamma+1}).$$

We consider a simple variance-exploding (VE) diffusion defined by the SDE

$$dx = dw. \tag{22}$$

The DDIM sampler is a discretization of the reverse ODE

$$\frac{dx}{dt} = -\frac{1}{2}\nabla_x \log p_t(x),$$

and the DDPM sampler is a discretization of the reverse SDE

$$dx = -\nabla_x \log p_t(x)dt + d\overline{w}.$$

For $\mathsf{CFG_{DDIM}}$ or $\mathsf{CFG_{DDPM}}$, we replace the score with CFG score $\nabla_x \log p_{t,\gamma}(x)$.

At inference time we choose an initial sample $x_T \sim \mathcal{N}(0,T)$ and run $\mathsf{CFG_{DDIM}}$ from $t = T \to 0$ to obtain a final sample $x_0$. Note that the true distribution generated by the forward process in our setting is $p_T = \mathcal{N}(0, T+1)$, which becomes close to our inference-time terminal distribution $\mathcal{N}(0,T)$ for large $T$. Taking the limit of $T \to \infty$ in our setting thus corresponds to allowing the forward diffusion process to fully mix.

$\mathsf{CFG_{DDIM}}$  For Counterexample 1, the $\mathsf{CFG_{DDIM}}$ ODE has a closed-form solution (derivation in section A.5):

$$\mathsf{CFG_{DDIM}}: \quad \frac{dx}{dt} = -\frac{1}{2}\nabla_x \log p_{t,\gamma}(x)$$
$$= x_t \left( \frac{\gamma}{2(1+t)} + \frac{(1-\gamma)}{2(2+t)} \right)$$
$$\implies x_t = x_T \sqrt{\frac{(t+1)^{\gamma}(t+2)^{1-\gamma}}{(T+1)^{\gamma}(T+2)^{1-\gamma}}}.$$

That is, for a particular initial sample $x_T$, $\mathsf{CFG_{DDIM}}$ produces the sample $x_t$ at time $t$. Evaluating at $t = 0$ and taking the limit as $T \to \infty$ yields the ideal denoised $x_0$ sampled by $\mathsf{CFG_{DDIM}}$ given an initial sample $x_T$:

$$\widehat{x}_0^{\mathsf{CFG_{DDIM}}}(x_T) = x_T \sqrt{\frac{2^{1-\gamma}}{(T+1)^{\gamma}(T+2)^{1-\gamma}}}$$
$$\to x_T \sqrt{\frac{2^{1-\gamma}}{T}} \quad \text{as } T \to \infty.$$

To get the denoised distribution obtained by reverse-sampling with $\mathsf{CFG_{DDIM}}$, we need to average over the distribution of $x_T$:

$$\mathbb{E}_{x_T \sim \mathcal{N}(0,T)} [\widehat{x}_0^{\mathsf{CFG_{DDIM}}}(x_T)] = \mathcal{N}(0, T\frac{2^{1-\gamma}}{T}) = \mathcal{N}\left(0, 2^{1-\gamma}\right).$$

which is equation 11 in the main text.

**CFG$_{\text{DDPM}}$**   CFG$_{\text{DDPM}}$ also has a closed-form solution (derived in section A.5):

$$dx = -\nabla_x \log p_{t,\gamma}(x)dt + d\overline{w}$$

$$= x \left( \frac{\gamma}{(1+t)} + \frac{(1-\gamma)}{(2+t)} \right) dt + d\overline{w}$$

$$\implies x(t) = x_T \frac{(1+t)^\gamma (2+t)^{1-\gamma}}{(1+T)^\gamma (2+T)^{1-\gamma}} + (1+t)^\gamma (2+t)^{1-\gamma} \sqrt{\frac{1}{2\gamma-1}} \sqrt{\left( \frac{t+1}{t+2} \right)^{1-2\gamma} - \left( \frac{T+1}{T+2} \right)^{1-2\gamma}} \xi.$$

Similar to the CFG$_{\text{DDIM}}$ argument, we can obtain the final denoised distribution as follows:

$$\widehat{x}_0^{\text{CFG}_{\text{DDPM}}}(x_T) = x_T \frac{2^{1-\gamma}}{(1+T)^\gamma (2+T)^{1-\gamma}} + 2^{1-\gamma} \sqrt{\frac{1}{2\gamma-1}} \sqrt{2^{2\gamma-1} - \left( \frac{T+1}{T+2} \right)^{1-2\gamma}} \xi$$

$$\rightarrow x_T \frac{2^{1-\gamma}}{T} + \sqrt{\frac{2 - 2^{2-2\gamma}}{2\gamma-1}} \xi \quad \text{as } T \rightarrow \infty$$

$$\implies \mathop{\mathbb{E}}_{x_T \sim \mathcal{N}(0,T)} [\widehat{x}_0^{\text{CFG}_{\text{DDPM}}}(x_T)] = \mathcal{N}\left( 0, T \left( \frac{2^{1-\gamma}}{T} \right)^2 + \frac{2 - 2^{2-2\gamma}}{2\gamma-1} \right)$$

$$\rightarrow \mathcal{N}\left( 0, \frac{2 - 2^{2-2\gamma}}{2\gamma-1} \right),$$

which is equation 10 in the main text, and for $\gamma \gg 1$ becomes approximately

$$\mathop{\mathbb{E}}_{x_T \sim \mathcal{N}(0,T)} [\widehat{x}_0^{\text{CFG}_{\text{DDPM}}}(x_T)] \approx \mathcal{N}\left( 0, \frac{2}{2\gamma-1} \right).$$

In Figure 5, we confirm results (10, 11) empirically.

## A.2   COUNTEREXAMPLE 2

Counterexample 2 (9) is a Gaussian mixture with equal weights and variances.

$$c \in \{0, 1\}, \quad p(c=0) = \frac{1}{2}$$

$$p_0(x_0|c) \sim \mathcal{N}(\mu^{(c)}, 1), \quad \mu^{(0)} = -\mu, \quad \mu^{(1)} = \mu$$

$$p_0(x_0) \sim \frac{1}{2} p_0(x_0|c=0) + \frac{1}{2} p_0(x_0|c=1).$$

We noted in the main text that if $\mu$ is sufficiently large enough that the clusters are approximately disjoint, and $\gamma \geq 1$, then $p_{0,\gamma}(x|c) \approx p_0(x|c)$. To see this note that

$$p_0(x_0) \approx \frac{1}{2} p_0(x_0|0) \mathbb{1}_{x>0} + \frac{1}{2} p_0(x_0|1) \mathbb{1}_{x>0}$$

$$p_{0,\gamma}(x|c) \propto p_0(x|c)^\gamma p_0(x)^{1-\gamma}$$

$$= p_0(x) \left( \frac{p_0(x|c)}{p_0(x)} \right)^\gamma$$

$$\propto p_0(x) \left( \mathbb{1}_{\text{sign}(x)=\mu^{(c)}} \right)^\gamma$$

$$\approx p_0(x|c) \quad \text{for } \gamma \geq 1.$$

However, $p_{t,\gamma}(x|c) \neq p_t(x|c)$ since the noisy distributions do overlap/interact.

We don't have complete closed-form solutions for this problem like we did for Counterexample 1. We have the solution for conditional DDIM for the basic VE process $dx = dw$ (using the results from

the previous section):

$$\text{DDIM on } p_t(x|c): \quad \frac{dx}{dt} = -\frac{1}{2}\nabla_x \log p_t(x|c)$$

$$= -\frac{1}{2(1+t)}(\mu^{(c)} - x_t)$$

$$\implies x(t) = \mu^{(c)} + (x_T - \mu^{(c)})\sqrt{\frac{1+t}{1+T}},$$

but otherwise have to rely on empirical results. We do however have access to the ideal conditional and unconditional denoisers via the scores (Appendix A.6):

$$\nabla_x \log p_t(x|c) = -\frac{1}{2(1+t)}(\mu^{(c)} - x_t)$$

$$\nabla_x \log p_t(x) = \frac{\nabla_x p_t(x)}{p_t(x)} = \frac{\frac{1}{2}\sum_{c=0,1}\nabla_x p_t(x|c)}{p_t(x)}.$$

## A.3 COUNTEREXAMPLE 3

We consider a 3-cluster problem to investigate why $\text{CFG}_{\text{DDIM}}$ and $\text{CFG}_{\text{DDPM}}$ often appear similar in practice despite being different in theory. Counterexample 3 (9) is a Gaussian mixture with equal weights and variances. We vary the variance to investigate its effect on CFG.

$$c \in \{0, 1, 2\}, \quad p(c) = \frac{1}{3} \quad \forall c$$

$$p_0(x_0|c) \sim \mathcal{N}(\mu^{(c)}, \sigma), \quad \mu^{(0)} = -3, \quad \mu^{(1)} = 0, \quad \mu^{(2)} = 3$$

$$p_0(x_0) \sim \frac{1}{3}p_0(x_0|c=0) + \frac{1}{3}p_0(x_0|c=1) + \frac{1}{3}p_0(x_0|c=2).$$

We run $\text{CFG}_{\text{DDIM}}$ and $\text{CFG}_{\text{DDPM}}$ with $\gamma = 3$, for $\sigma = 1$ and $\sigma = 2$. Results are shown in Figure 5.

## A.4 GENERALIZATION EXAMPLE 4

We consider a multi-cluster problem to explore the impact of guidance on generalization:

$$p_0(x) \sim \mathcal{N}(0, 10)$$

$$p_0(x|c=0) \sim \sum_i w_i \mathcal{N}(\mu_i, \sigma) \tag{23}$$

$$\mu = (-3, -2.5, -2, -1.5, -1, -0.5, 0, 0.5, 1, 1.5, 2, 2.5)$$

$$w_i = 0.0476 \quad \forall i \neq 6; \quad w_6 = 0.476$$

$$\sigma = 0.1$$

Note that the unconditional distribution is wide enough to be essentially uniform within the numerical support of the conditional distribution. The conditional distribution is a GMM with evenly spaced clusters of equal variance, and all equal weights, except for a "dominant" cluster in the middle with higher weight. The results are shown in Figure 6.

## A.5 CLOSED-FORM ODE/SDE SOLUTIONS

First, we want to solve equations of the general form $\frac{dx}{dt} = -a(t)x + b(t)$, which will encompass the ODEs and SDEs of interest to us. All we need for the ODEs is the special $b(t) = a(t)c$, which is easier.

The main results are

$$\frac{dx}{dt} = a(t)(c - x)$$

$$\implies x(t) = c + (x_T - c)e^{A(T)-A(t)} \tag{24}$$

$$\text{where } A(t) = \int a(t)dt$$

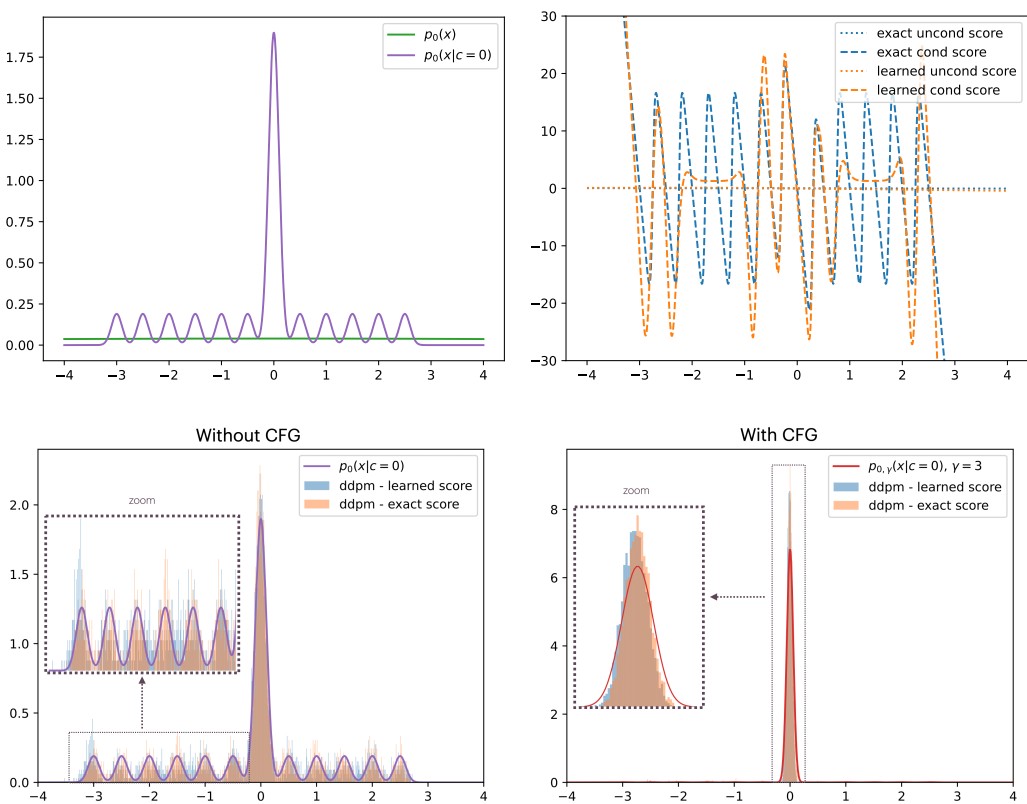

Figure 6: **An example where guidance benefits generalization.** (Top left) Conditional $p_0(x|c=0)$ (purple) and unconditional $p_0(x)$ (green) distributions for Example 4 (equation 23). The unconditional distribution is approximately uniform, while the conditional distribution for $c = 0$ is a GMM with several clusters with equal variances, and equal weights except for a single "dominant" cluster with a higher weight. (Top right) We train small MLPs to predict the conditional and unconditional scores, with early-stopping so that the fit is imperfect. We plot the exact (orange) vs. learned (blue) conditional and unconditional scores: the unconditional scores are learned accurately, while the conditional scores are learned accurately near the dominant cluster but poorly elsewhere. (Bottom left) We sample with DDPM on the conditional distribution (no guidance) using learned scores (blue) vs. exact scores (orange). We expect DDPM to generate the conditional distribution $p_0(x|c=0)$ (purple). However, DDPM-with-learned-scores samples less accurately than DDPM-with-exact-scores away from the dominant cluster (where the learned scores are inaccurate) (compare the increased blue vs. orange sampling in low-probability regions). (Bottom right) With guidance $\gamma = 3$, $p_{0,\gamma}(x|c)$ (red) and both samplers concentrate around the dominant cluster (where the learned scores are accurate), reducing the generalization gap between the learned and exact models.

and

$$\frac{dx}{dt} = -a(t)x + b(t)$$

$$\implies x(t) = e^{-A(t)}(B(t) - B(T)) + x_T e^{A(T) - A(t)} \tag{25}$$

$$\text{where } A(t) = \int a(t)dt, \quad B(t) = \int e^{A(t)}b(t)dt.$$

First let's consider the special case $b(t) = a(t)c$, which is easier. We can solve it (formally) by separable equations:

$$\frac{dx}{dt} = a(t)(c - x)$$

$$\implies \int \frac{1}{c - x}dx = \int a(t)dt = A(t)$$

$$\implies -\log(c - x) = A(t) + C$$

$$\implies c - x = e^{-A(t) - C}$$

$$\implies x(t) = c + Ce^{-A(t)}. \tag{26}$$

Next we need to apply initial conditions to get the right constants. Remembering that we are actually sampling backward in time from initialization $x_T$, we can solve for the constant $C$ as follows, to obtain result (24):

$$x_T = c + Ce^{-A(T)}$$

$$\implies C = e^{A(T)}(x_T - c)$$

$$\implies x(t) = c + (x_T - c)e^{A(T) - A(t)}.$$

We will apply this result to $\mathsf{CFG}_{\mathsf{DDIM}}$ shortly, but for now we note that for a VE diffusion $dx = \sqrt{t}dw$ on a Gaussian data distribution $p_0(x) \sim \mathcal{N}(\mu, \sigma)$ the above result implies the exact DDIM dynamics:

$$p_t(x) \sim \mathcal{N}(\mu, \sigma^2 + t)$$

$$\text{DDIM on } p_t(x): \frac{dx}{dt} = -\frac{1}{2}\nabla_x \log p_t(x)$$

$$= -\frac{1}{2(\sigma^2 + t)}(\mu - x)$$

$$A(t) = -\frac{1}{2}\log(\sigma^2 + t)$$

$$\implies x_t = \mu + (x_T - \mu)e^{A(T) - A(t)}$$

$$= \mu + (x_T - \mu)\sqrt{\frac{\sigma^2 + t}{\sigma^2 + T}}.$$

(which makes sense since $x_{t=T} = x_T$ and $\frac{\sqrt{\sigma^2}}{\sqrt{\sigma^2 + T}} \approx 0 \implies x_{t=0} \approx \mu$).

Now let's return to the general problem with arbitrary $b(t)$ (we need this for the SDEs). We can use an integrating factor to get a formal solution:

$$\frac{dx}{dt} = -a(t)x + b(t)$$

$$\text{Integrating factor: } e^{A(t)}, \quad A(t) = \int a(t)dt$$

$$\frac{d}{dt}(x(t)e^{A(t)}) = (x'(t) + a(t)x(t))\,e^{A(t)}$$

$$= b(t)e^{A(t)}$$

$$\implies e^{A(t)}x(t) = \int e^{A(t)}b(t)dt + C$$

$$\implies x(t) = e^{-A(t)}\int e^{A(t)}b(t)dt + Ce^{-A(t)}. \tag{27}$$

Note that if $b(t) = a(t)c$ this reduces to (26):

$$\int e^{-A(t)} e^{A(t)} b(t) dt = c e^{-A(t)} \int a(t) e^{A(t)} dt = c$$

$$\implies x(t) = c + C e^{-A(t)}.$$

Again, we need to apply boundary conditions to get the constant, and remember that we are actually sampling backward in time from initialization $x_T$ to obtain result (25):

$$\frac{dx}{dt} = -a(t)x + b(t)$$

$$x_T = e^{-A(T)} B(T) + C e^{-A(T)}, \quad B(t) := \int e^{A(t)} b(t) dt$$

$$\implies C = e^{A(T)} x_T - B(T)$$

$$\implies x(t) = e^{-A(t)} B(t) + (e^{A(T)} x_T - B(T)) e^{-A(t)}$$

$$= e^{-A(t)}(B(t) - B(T)) + x_T e^{A(T)-A(t)}.$$

Note that for $b(t) = a(t)c$ this reduces (24):

$$b(t) = a(t)c \implies B(t) = c e^{A(t)}$$

$$\implies x(t) = -c e^{-A(t)} (e^{A(t)} - e^{A(T)}) + x_T e^{A(T)-A(t)}$$

$$= c + (x_T - c) e^{A(T)-A(t)}.$$

**Counterexample 1 solutions** To solve the $\mathsf{CFG_{DDIM}}$ ODE for Counterexample 1 (Equation 9) we apply result (24):

$$\frac{dx}{dt} = a(t)(c - x) \implies x(t) = c + (x_T - c) e^{A(T)-A(t)}$$

$$a(t) = -\frac{\gamma}{2(1+t)} - \frac{(1-\gamma)}{2(2+t)}, \quad c = 0$$

$$A(t) = -\frac{1}{2} \int \frac{\gamma}{(1+t)} + \frac{(1-\gamma)}{(2+t)} dt$$

$$= -\frac{1}{2}(\gamma \log(t+1) + (\gamma - 1)\log(t+2))$$

$$\implies x_t = x_T \sqrt{\frac{(t+1)^\gamma (t+2)^{1-\gamma}}{(T+1)^\gamma (T+2)^{1-\gamma}}}.$$

To solve the $\mathsf{CFG_{DDPM}}$ SDE for Counterexample 1 (Equation 9), we first apply (25) to the SDE with $b(t) = -\xi(t)$:

$$\frac{dx}{dt} = -a(t)x - \xi(t), \quad \langle \xi(t) \rangle = 0, \quad \langle \xi(t), \xi(t') \rangle = \delta(t - t')$$

$$\implies x(t) = x_T e^{A(T)-A(t)} + e^{-A(t)}(B(t) - B(T)), \quad A(t) = \int a(t) dt, \quad B(t) = -\int e^{A(t)} \xi(t) dt$$

$$= x_T e^{A(T)-A(t)} + e^{-A(t)} \sqrt{\int_t^T e^{2A(t)} dt} \xi.$$

Now, plugging in the DDPM drift term we find that

$$a(t) = -\frac{\gamma}{(1+t)} - \frac{(1-\gamma)}{(2+t)}$$

$$A(t) = -\gamma \log(1+t) - (1-\gamma)\log(2+t)$$

$$e^{A(t)} = (1+t)^{-\gamma}(2+t)^{-1+\gamma}$$

$$\int e^{2A(t)}dt = \int (1+t)^{-2\gamma}(2+t)^{-2+2\gamma}dt$$

$$= -\frac{1}{2\gamma - 1}\left(\frac{t+1}{t+2}\right)^{1-2\gamma}$$

$$x(t) = x_T e^{A(T)-A(t)} + e^{-A(t)}\sqrt{\int_t^T e^{2A(t)}dt}\,\xi$$

$$= x_T \frac{(1+t)^{\gamma}(2+t)^{1-\gamma}}{(1+T)^{\gamma}(2+T)^{1-\gamma}} + (1+t)^{\gamma}(2+t)^{1-\gamma}\sqrt{\frac{1}{2\gamma - 1}}\sqrt{\left(\frac{t+1}{t+2}\right)^{1-2\gamma} - \left(\frac{T+1}{T+2}\right)^{1-2\gamma}}\,\xi.$$

### A.6 EXACT DENOISER FOR GMM

For the experiments in Figure 2, we used an exact denoiser, for which we require exact conditional and unconditional scores. Exact scores are available for any GMM as follows. This is well-known (e.g. Karras et al. (2024a)) but repeated here for convenience.

$$p(x) = \sum w_i \phi(x; \mu_i, \sigma_i), \quad \text{where} \quad \phi(x; \mu, \sigma^2) := \frac{1}{\sqrt{2\pi}\sigma}e^{-\frac{(x-\mu)^2}{2\sigma^2}}$$

$$\implies \nabla \log p(x) = \frac{\nabla p(x)}{p(x)}$$

$$= \frac{\sum w_i \nabla \phi(\mu_i, \sigma_i)}{\sum w_i \phi(\mu_i, \sigma_i)}$$

$$= -\frac{\sum w_i \left(\frac{x-\mu_i}{\sigma_i^2}\right)\phi(x; \mu_i, \sigma_i^2)}{\sum w_i \phi(\mu_i, \sigma_i)}.$$

## B PCG SDE

We want to show that the SDE limit of Algorithm 1 with $K = 1$ is

$$dx = \Delta\mathsf{DDIM}(x,t) + \Delta\mathsf{LD}_\mathsf{G}(x,t,\gamma).$$

To see this, note that a single iteration of Algorithm 1 with $K = 1$ expands to

$$x_t = x_{t+\Delta t} \underbrace{-\frac{1}{2}\beta_t(x_{t+\Delta t} - \nabla \log p_{t+\Delta t}(x_{t+\Delta t}|c))\Delta t}_{\text{DDIM step on } p_{t+\Delta t}(x+\Delta t|c)} + \underbrace{\frac{\beta_t \Delta t}{2}\nabla \log p_{t,\gamma}(x_t|c) + \sqrt{\beta_t \Delta t}\mathcal{N}(0, I_d)}_{\text{Langevin dynamics on } p_{t,\gamma}(x|c)}$$

$$\implies dx = \lim_{\Delta t \to 0} x_t - x_{t+\Delta t} = \underbrace{-\frac{1}{2}\beta_t(x_t - \nabla \log p_t(x_t|c))dt}_{\Delta\mathsf{DDIM}(x,t)} + \underbrace{\frac{1}{2}\beta_t \nabla \log p_{t,\gamma}(x_t|c)dt + \sqrt{\beta_t}d\overline{w}}_{\Delta\mathsf{LD}_\mathsf{G}(x,t,\gamma)}.$$

This concludes the proof.

A subtle point in the argument above is that $\Delta\mathsf{LD}_\mathsf{G}(x,t,\gamma)$ represents the result of the Langevin step in the PCG corrector update, rather than the differential of an SDE. In Algorithm 1, $t$ remains constant during the LD iteration, and so the SDE corresponding to the LD iteration is

$$dx = \frac{1}{2}\beta_t \nabla \log p_{t,\gamma}(x_t|c)ds + \sqrt{\beta_t}d\overline{w}, \tag{28}$$

Increasing # Langevin Steps (PCG_DDPM) $\longrightarrow$

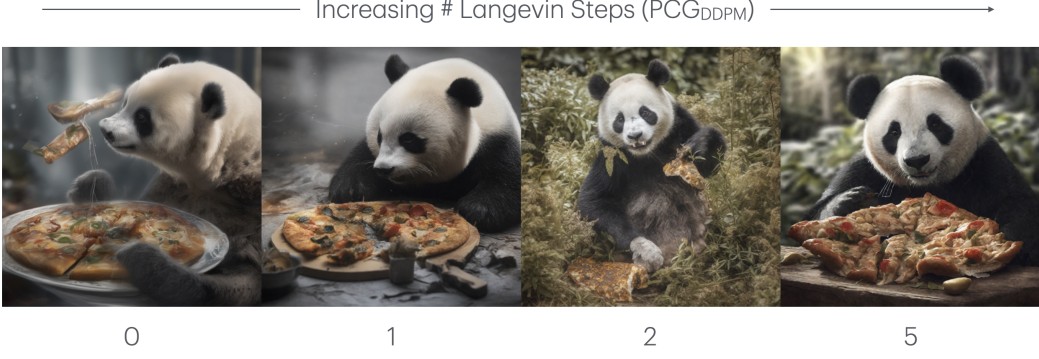

| 0 | 1 | 2 | 5 |

Figure 7: **Effect of Langevin Dynamics**. PCG generations with $\gamma = 1$ (no guidance) fixed and number of Langevin steps $K$ varied. The prompt is "photograph of a panda eating pizza". Increasing the number of Langevin steps can qualitatively improve image quality, even without guidance.

where $s$ is an LD time-axis that is distinct from the denoising time $t$, which is fixed during the LD iteration. Thus $\Delta \mathsf{LD}_\mathsf{G}(x, t, \gamma)$ is not the differential of (28) (the difference is $dt$ vs $ds$). However, when we take an LD step of length $dt$ as required for the PCG corrector, the result is

$$\int_0^{dt} -\frac{\beta_t}{2}\nabla \log p_{t,\gamma} ds + \sqrt{\beta_t} d\overline{w} = -\frac{\beta_t}{2}\nabla \log p_{t,\gamma} dt + \sqrt{\beta_t} d\overline{w} = \Delta \mathsf{LD}_\mathsf{G}(x, t, \gamma),$$

so $\Delta \mathsf{LD}_\mathsf{G}(x, t, \gamma)$ represents the result of the PCG corrector update in the limit as $\Delta t \to 0$.

## C  ADDITIONAL SAMPLES AND METRICS

Table 2: FD-DINOv2 scores for PCG, DDIM, and PCG over $\gamma$ and number of LD steps. Setup as described in Table 1.

| Method | $\gamma = 1$ | $\gamma = 1.1$ | $\gamma = 1.3$ | $\gamma = 1.5$ |
|---|---|---|---|---|
| DDPM-CFG | 161.72 | 125.71 | 84.65 | 65.44 |
| DDIM-CFG | 189.76 | 152.04 | 104.17 | 79.07 |
| PCG LD steps = 1 | 188.83 | 155.19 | 109.11 | 83.50 |
| PCG LD steps = 3 | 174.97 | 119.87 | 73.38 | 70.80 |
| PCG LD steps = 5 | 166.38 | 110.27 | 71.08 | 93.21 |

Table 3: Inception Scores for PCG, DDIM, and PCG over $\gamma$ and number of LD steps. Setup as described in Table 1.

| Method | $\gamma = 1$ | $\gamma = 1.1$ | $\gamma = 1.3$ | $\gamma = 1.5$ |
|---|---|---|---|---|
| DDPM-CFG | 108.2628 | 126.8507 | 157.0371 | 178.0676 |
| DDIM-CFG | 100.0823 | 116.3814 | 144.7761 | 164.6486 |
| PCG LD steps = 1 | 101.2306 | 113.6755 | 133.1969 | 147.5756 |
| PCG LD steps = 3 | 105.2118 | 126.9752 | 152.2398 | 160.9198 |
| PCG LD steps = 5 | 107.1457 | 139.8954 | 155.7239 | 149.6180 |

## D  AN ALTERNATIVE DISCRETIZATION

In this section we empirically study an alternative discretization of PCG. The equivalence between PCG and CFG holds in the SDE limit as $\Delta t \to 0$, so PCG should be thought of as an SDE for which Algorithm 1 is one choice of discretization. However, other discretizations are possible. In this section we explore one of these. In particular, we make a single change to Algorithm 1: we modify

the LD loop by changing the order of operations: we first add noise, and then compute and step in the direction of the score; specifically, the inner loop LD becomes:

$$
\begin{aligned}
x_t &\leftarrow x_t + \sqrt{\varepsilon}\eta, \quad \eta \sim \mathcal{N}(0, I_d) \\
s_{t,\gamma} &:= (1-\gamma)\nabla \log p_t(x_t) + \gamma \nabla \log p_t(x_t|c) \\
x_t &\leftarrow x_t + \frac{\varepsilon}{2} s_{t,\gamma}
\end{aligned}
\tag{29}
$$

This is similar to the "churn" operation in Karras et al. (2022)'s stochastic sampler, and conceptually similar to a noise-then-denoise step in Lugmayr et al. (2022). We generally find that this change improves the PCG metrics (more closely matching the DDPM metrics) for smaller $\gamma$'s, while worsening the metrics for larger $\gamma$'s, as shown in Table 4. We are not sure why this is, but it is well-known that diffusion models are sensitive to discretization choices in practice.

Table 4: Metrics for DDPM, DDIM, and PCG over $\gamma$ and number of LD steps. Alternative LD discretization (Equation 29).

| FID | $\gamma = 1$ | $\gamma = 1.1$ | $\gamma = 1.3$ | $\gamma = 1.5$ |
|---|---|---|---|---|
| PCG LD steps = 1 | 5.87115 | 4.72043 | 4.15484 | 4.74044 |
| PCG LD steps = 3 | 4.79793 | 3.49296 | 4.82135 | 7.69348 |
| PCG LD steps = 5 | 4.51476 | 3.35029 | 6.04134 | 10.6716 |
| FD-DINOv2 | $\gamma = 1$ | $\gamma = 1.1$ | $\gamma = 1.3$ | $\gamma = 1.5$ |
| PCG LD steps = 1 | 156.854 | 132.605 | 102.107 | 88.2433 |
| PCG LD steps = 3 | 137.502 | 100.912 | 76.9214 | 86.1473 |
| PCG LD steps = 5 | 129.782 | 89.3722 | 79.0756 | 112.229 |
| Inception Score | $\gamma = 1$ | $\gamma = 1.1$ | $\gamma = 1.3$ | $\gamma = 1.5$ |
| PCG LD steps = 1 | 107.7871 | 117.3694 | 132.3872 | 141.6556 |
| PCG LD steps = 3 | 115.4412 | 131.1285 | 148.9654 | 152.2574 |
| PCG LD steps = 5 | 117.5658 | 136.5819 | 150.1884 | 138.9601 |

# E  ALGORITHMS

Algorithm 2 provides an explicit, practical implementation of PCG. Note that Algorithm 1 and 2 have slightly different DDIM steps, but this just corresponds to two different discretizations of the same process. Algorithm 1 uses the first-order Euler–Maruyama discretization known as "reverse SDE" (Song et al., 2020), which is convenient for our mathematical analysis. Algorithm 2 uses the original DDIM discretization (Song et al., 2021), equivalent to a more sophisticated integrator (Lu et al., 2022a), which is more common in practice.

---

**Algorithm 2:** $\mathsf{PCG}_{\mathsf{DDIM}}$, explicit

**Input:** Conditioning $c$, guidance weight $\gamma \geq 0$
**Constants:** $\{\alpha_t\}, \{\overline{\alpha}_t\}, \{\beta_t\}$ from Ho et al. (2020)

1  $x_1 \sim \mathcal{N}(0, I)$
2  **for** $(t = 1 - \Delta t;\ t \geq 0;\ t \leftarrow t - \Delta t)$ **do**
3  $\quad \varepsilon, \varepsilon_c := \mathsf{NoisePredictionModel}(x_{t+\Delta t}, c)$
4  $\quad \widehat{x}_0 := (x_{t+\Delta t} - \sqrt{1 - \overline{\alpha}_{t+\Delta t}}\varepsilon_c)/\sqrt{\overline{\alpha}_{t+\Delta t}}$
5  $\quad x_t := \sqrt{\overline{\alpha}_t}\widehat{x}_0 + \sqrt{1 - \overline{\alpha}_t}\varepsilon_c$  $\qquad\qquad \triangleright$ DDIM step $p_{t+\Delta t}(x|c) \rightarrow p_t(x|c)$
6  $\quad$ **for** $k = 1, \ldots K$ **do**
7  $\quad\quad \varepsilon, \varepsilon_c := \mathsf{NoisePredictionModel}(x_t, c)$
8  $\quad\quad x_t \leftarrow x_t - \frac{\beta_t}{2\sqrt{1 - \overline{\alpha}_t}}\left((1-\gamma)\varepsilon + \gamma\varepsilon_c\right) + \sqrt{\beta_t}\eta$  $\quad \triangleright$ Langevin dynamics on $p_{t,\gamma}(x|c)$
9  $\quad$ **end**
10  **end**
11  **return** $x_0$

---

