# OpenReview forum: "Classifier-Free Guidance is a Predictor-Corrector"
_ICLR.cc/2025/Conference — Submitted to ICLR 2025_

### Official Review · Reviewer_CU4e · 2024-10-18

**Soundness:** 3
**Presentation:** 2
**Contribution:** 2
**Rating:** 5
**Confidence:** 4

**Summary:**

This paper focuses on the theoretical understanding of classifier-free guidance (CFG), a widely used technique in conditional sampling with diffusion models. The authors argue that the theory of CFG has been somewhat misunderstood, presenting counterexamples using 1D toy models to support their claim. They show that CFG can be explained by a predictor-corrector (PC) sampling algorithm with different annealing distributions. In particular, they introduce the predictor-corrector guidance (PCG) and suggest that CFG with DDPM sampling is equivalent to PCG with DDIM sampling. In this framework, the predictor is set as DDIM and the corrector is set as Langevin dynamics with a gamma-powered distribution.

**Strengths:**

* The paper points out that the theoretical understanding of CFG is lacking, considering its widespread practical use. The analysis of how different distributions correspond to different guidance scales could be helpful for practical applications.

* The statement of CFG with DDPM through predictor-corrector sampling, where Langevin dynamics serve as the corrector, is intuitive and reasonable.

**Weaknesses:**

* The paper provides only informal theorems, so it is unclear what specific statements the authors intend to make within the scope of their work.
  * The theorems are incomplete and difficult to fully understand. In particular, the notation used in the statements is not well-defined (e.g., what is meant by c=0?), and the assumptions necessary to satisfy these theorems are not properly discussed.
  * Specifically, in my opinion, the additional claim in Theorem 1 that the DDIM variant is exponentially sharper than the DDPM variant is based solely on the counterexamples, which may lead to an overstatement in its current form.
  * For Theorem 3, the analysis only covers CFG with DDPM, so a clearer statement regarding this limitation is needed.

* While the relationship between CFG and PCG is explained, the reasons why CFG works are not adequately addressed.
  * There is a lack of sufficient analysis regarding PCG. As the authors themselves note, unlike the conventional PC algorithm, PCG operates with different annealing distributions for the predictor and corrector. Thus, the effectiveness of PCG should be explained with an analysis based on these different annealing distributions. For example, the effect of different annealing distributions on the final sampled distribution is not discussed. I believe this analysis is crucial because it ties into the argument that CFG works with a sampling distribution that deviates from the conventional intuition.
  * In explaining CFG in terms of PCG, the authors assume that the difference in timesteps between the predictor and corrector tends to zero, but the implications of this assumption are not sufficiently discussed or analyzed.
  * In Line 465, the paper mentions that CFG and PCG are qualitatively similar and claims that the results are consistent with the theory. However, looking at the quantitative metrics in Table 1, there appears to be a difference, so I question whether this statement is valid.

* CFG is known to be effective for image-condition alignment. It would be beneficial to include experimental results, such as quantitative metrics for image-text alignment in text-to-image diffusion models such as Stable Diffusion.

**Questions:**

* Please provide the authors' responses to the points listed under "Weaknesses".
* Algorithm 2 states that the noise prediction model uses the same timestep for both the DDIM step and the Langevin dynamics step. Is this correct?
* Minor comments
  * I believe that Eq. 2 should be expressed as a proportional relationship.
  * In Line 131, the paper states that it primarily considers the VP diffusion process, but the counterexamples seem to primarily focus on the VE diffusion process.

---

> ### Author Response · Authors · 2024-11-15
>
> We thank the reviewer for their careful review of our paper and helpful feedback. The reviewer clearly put significant effort into a detailed reading and thoughtful questions and comments, and we truly appreciate this.
>
> We will address each point that was raised individually. We also updated the PDF based on your feedback; briefly, the main changes are:
>
> * We added a fully formal statement and proof of our claims, as Theorem 4 in the Appendix. We also edited the wording around Theorems 1 and 2 in the body, to clarify that they are informal statements (meant to convey the main intuitions of our formal claims).
> * We added three remarks in Section 4.1, discussing the relations between PCG and annealed Langevin dynamics more explicitly.
> * We fixed typos in Algorithms 1 & 2 which cause misunderstanding.
> * We added several exploratory experiments considering the effect of different discretization choices (Appendix D, Table 4).
>
> Please let us know if you have additional questions or concerns; we are happy to elaborate further.

---

> ### Author Response · Authors · 2024-11-15
> **Responses to Weaknesses (Part 1)**
>
> (Original questions in **bold**)
>
> * **The paper provides only informal theorems, so it is unclear what specific statements the authors intend to make within the scope of their work.**
>     * **The theorems are incomplete and difficult to fully understand. In particular, the notation used in the statements is not well-defined (e.g., what is meant by c=0?), and the assumptions necessary to satisfy these theorems are not properly discussed.**
>         * Thanks for identifying this source of confusion. We initially stated these theorems informally in order to convey the main intuitions. However, we have now added Theorem 4 in the Appendix with a formal and mathematically-self-contained statement and proof. We also edited Theorems 1 and 2 and the surrounding text to clarify the claims, and avoid misunderstanding these as formal claims.
>     * **Specifically, in my opinion, the additional claim in Theorem 1 that the DDIM variant is exponentially sharper than the DDPM variant is based solely on the counterexamples, which may lead to an overstatement in its current form.**
>         * Apologies, we did not intend to claim that DDIM-CFG is always exponentially sharper than DDPM; only that this is so for our particular counterexample (hence our theorem statement that “there exists...”). Our edits aim to clarify this.
>     * **For Theorem 3, the analysis only covers CFG with DDPM, so a clearer statement regarding this limitation is needed.**
>         * We do mention the limitation that our analysis only applies to CFG with DDPM near the beginning of section 5.2, but we also added some extra clarifying text around Theorem 3 to emphasize this.

---

> ### Author Response · Authors · 2024-11-15
> **Responses to Weaknesses (Part 2)**
>
> * **While the relationship between CFG and PCG is explained, the reasons why CFG works are not adequately addressed.**
>     * **There is a lack of sufficient analysis regarding PCG. As the authors themselves note, unlike the conventional PC algorithm, PCG operates with different annealing distributions for the predictor and corrector. Thus, the effectiveness of PCG should be explained with an analysis based on these different annealing distributions. For example, the effect of different annealing distributions on the final sampled distribution is not discussed. I believe this analysis is crucial because it ties into the argument that CFG works with a sampling distribution that deviates from the conventional intuition.**
>         * Good question; we added several Remarks in Section 4 to clarify this (Remarks 1, 2, and 3). Essentially, PCG can be thought of as an annealed Langevin dynamics, which in general does not requires the annealing and LD distributions to “match”. In fact, annealing simply by reducing the temperature without changing the distribution at all can work, as long as it allows the LD to mix. The main insight we borrow from Song’s predictor-corrector method is that the reverse diffusion process offers a natural and effective annealing schedule which can enable successful mixing (even if the annealed distributions do not exactly match the distribution LD attempts to sample from).
>     * **In explaining CFG in terms of PCG, the authors assume that the difference in timesteps between the predictor and corrector tends to zero, but the implications of this assumption are not sufficiently discussed or analyzed.**
>         * In our theoretical analysis, we show the equivalence between CFG and PCG in the SDE limit as $dt \to 0$. In that case, the difference in timesteps between the predictor and corrector does tend to zero. It is analogous to the previously-known equivalence between DDPM and DDIM+LD, which only holds in the continuous-time limit.
>         * Of course, as you point out, discretization choices are important in practice. Algorithm 1 is one possible discretization of the PCG SDE, in which the predictor step takes us from time $t + dt$ to time $t$, and the corrector step acts at time $t$. However, other discretizations are also possible and may be beneficial (please see next question).
>         * In case it’s helpful, here is another way to think about the PCG algorithm. If you consider the entire sequence $\ldots \text{Predictor}(t+dt) \to \text{Corrector}(t) \to \text{Predictor}(t) \to \text{Corrector}(t - dt) \ldots$, then whether you think of this as steps of $(C_t, P_t)$ or $(P_{t+dt}, C_t)$ is just an analysis detail, i.e. the following two yield the same overall sequence:
>             * Corrector, then Predictor, using the same timestep $t$
>             * Predictor at $t+dt$, then Corrector at $t$
>     * **In Line 465, the paper mentions that CFG and PCG are qualitatively similar and claims that the results are consistent with the theory. However, looking at the quantitative metrics in Table 1, there appears to be a difference, so I question whether this statement is valid.**
>         * While we have proven that CFG and PCG are equivalent in the SDE limit, discretization choices are known to be very important in practice for diffusion in general. For example, even standard DDPM without CFG improves by increasing the number of steps, and popular solvers like DPM++ (Lu, 2023) rely on careful discretization. Carefully tuning the discretization and other practical parameters of PCG was outside the scope of this work; we only aimed to show the equivalence theoretically and provide empirical evidence of its plausibility.
>         * That said, we appreciate your concern about the gap in metrics between PCG and DDPM-CFG, particularly for small $\gamma$. We wondered about this too and suspected it was due to discretization. To explore this, we have added another set of experiments with an alternative choice of discretization of the PCG SDE (Table 4). With the alternative discretization, the metrics generally improve for small $\gamma$ (1 or 1.1), and more closely match DDPM-CFG (especially for $\gamma=1$, where there was a significant gap with the original discretization).  However, for larger $\gamma$ the original discretization still yields better metrics — we are not sure why this is, but it highlights the sensitivity of the results to discretization and other implementation choices.
>
> * **CFG is known to be effective for image-condition alignment. It would be beneficial to include experimental results, such as quantitative metrics for image-text alignment in text-to-image diffusion models such as Stable Diffusion.**
>     * Thank you, we agree this is a good idea. For example, we could measure CLIP-scores for CFG vs PCG. We will consider adding these experiments for the camera-ready.

---

> ### Author Response · Authors · 2024-11-15
> **Responses to Questions (Part 3)**
>
> * **Algorithm 2 states that the noise prediction model uses the same timestep for both the DDIM step and the Langevin dynamics step. Is this correct?**
>     * Thank you for noticing this. We had a typo and an unclear comment In Algorithm 2, which we have fixed in the updated pdf. We actually do update the timestep between DDIM and LD (and then it remains fixed within the LD loop). For $\gamma=1$, the DDIM step denoises from $p_{t+dt}$ to $p_t$, and then LD runs on the distribution $p_t$.  Our code implementation follows the algorithm as stated. Please let us know if the updated pdf is still unclear.
>
> * **Minor comments**
>     * **I believe that Eq. 2 should be expressed as a proportional relationship.**
>         * Yes, thanks!
>     * **In Line 131, the paper states that it primarily considers the VP diffusion process, but the counterexamples seem to primarily focus on the VE diffusion process.**
>         * Good point; we edited line 131 to remove the misleading statement. The counterexamples do indeed use the VE process.

---

> > ### Comment · Reviewer_CU4e · 2024-11-26
> >
> > Thank you for the authors' response. I have reviewed the revised manuscript and see significant improvements over the previous version. Since some concerns have been addressed, I would like to increase my score from 3 to 5. However, I still have concerns and would like to leave the following comments:
> >
> > * I believe that the formal version of the theorem, Theorem 4 in the revised appendix, still does not clearly express the intended statement. The definition, statement, and proof are mixed together and not formulated in a theoretical way. In my opinion, theorem statements should provide a clear and concise description.
> >
> > * While I understand that the annealing distributions of the predictor and corrector can be different, I believe that there should be more discussion regarding the implications of these settings in PCG. This discussion would highlight the strengths of PCG.
> >
> > * The lack of application of CFG in widely used text-to-image models still limits the strengths of this method from being fully emphasized.

---

> > > ### Author Response · Authors · 2024-11-27
> > >
> > > Point 1.
> > > We agree with the reviewer that our initial statement of Theorem 4 was too long and mixed definitions with the statement itself, harming overall clarity. We have posted a revised draft where we split it into two definition and a concise theorem statement. We appreciate the feedback and hope our changes help!
> > >
> > > Point 2.
> > > Regarding the difference between the predictor and corrector distributions, in our first revision we added Remarks 1-3 on page 6 to help clarify their respective roles. Specifically, the DDIM predictor provides a good annealing path, while the LD corrector samples from a sharpened distribution (in fact, with enough LD steps, we would sample exactly from the $p_{0, \gamma}$ at time $t=0$ — the very thing CFG is “supposed” to do but does not quite achieve).  Also, in Figure 4 we explore the PCG design space with SDXL. We vary the guidance strength $\gamma$ and number of LD steps $K$, which adjust the corrector in different ways, with different qualitative effects on prompt-adherence and image quality. If we haven’t correctly understood your question about the “implications of these settings in PCG”, could you please clarify which aspect(s) you’d like to discuss further?
> > >
> > > Point 3.
> > > Fair. Our main goal in this work was understanding CFG rather than proposing a new method (as our main result is the *equivalence* between CFG and a particular form of PCG, we are not claiming that PCG is better — although it exposes a design space that *could* be better). Our ImageNet and SDXL experiments were meant to confirm this equivalence and do a preliminary exploration of the design space, but there is certainly room for future work in this area.
> > >
> > > Again, we’re grateful for your feedback and help in improving our paper. Your points with regard to the counterexample theorems were especially helpful. If you believe the revised paper is of sufficient quality and value to appear at ICLR, we kindly ask that you consider raising your score to a “weak accept”.

---

### Official Review · Reviewer_q66d · 2024-10-29

**Soundness:** 2
**Presentation:** 3
**Contribution:** 2
**Rating:** 6
**Confidence:** 3

**Summary:**

The paper aims to investigate the theoretical foundations of classifier-free guidance (CFG). It disproves common misconcepts by  using counterexamples to show that CFG does not generate gamma-powered distribution, and CFG interacts differently with DDPM and DDIM. The  paper shows that CFG is equivalent to a particular kind of predictor-corrector that combines one step of DDIM denoiser with one step of Langevin dynamics in the gamma-powered distribution.

**Strengths:**

1. The paper disproves the misconcepts about CFG using counterexamples.
2. The paper provides a new understanding of CFG from the perspective of predictor-corrector guidance.

**Weaknesses:**

1. In page 3, the authors state that `` This gives a principled way to interpret CFG: it is implicitly an annealed
Langevin dynamics''.  What is the exact annealing path of the associated annealed Langevin dynamics? It seems not clear to me that CFG can be directly associated with  annealed Langevin dynamics as the predictor and corrector correspond to different limiting distributions and the corrector take only one Langevin dynamics.

2. The interpretations of Theorem 1 and 2 are not clear stated. Is CFG-DDIM always tends to be sharper than CFG-DDPM, or it just because the special construction used in Theorem 1 and 2?

3. What is the potential usefulness of the derived results in further theoretical analysis of diffusion model?

**Questions:**

Is it always true that  a larger $\gamma$ and more Langevin dynamic steps in the corrector can lead to sharper distribution?

---

> ### Author Response · Authors · 2024-11-15
> **Responses to Weaknesses and Questions**
>
> We thank the reviewer for their careful review of our paper and helpful feedback. We address each point that was raised individually (original questions **bolded**). We also updated the PDF based on your feedback and that of other reviewers as detailed in the *Response to all reviewers*
>
> **Weaknesses:**
>
> **In page 3, the authors state that `` This gives a principled way to interpret CFG: it is implicitly an annealed Langevin dynamics''. What is the exact annealing path of the associated annealed Langevin dynamics? It seems not clear to me that CFG can be directly associated with annealed Langevin dynamics as the predictor and corrector correspond to different limiting distributions and the corrector take only one Langevin dynamics.**
>
> * [A similar question was asked by reviewer CU4e; our response here is similar] We added several Remarks in Section 4 to to help clarify this (Remarks 1, 2, and 3). PCG is an annealed Langevin dynamics where the annealing path is given by the reverse diffusion process and the LD operates on the gamma-powered distribution $p_{t, \gamma}$. Note that in general, annealed LD does not requires the annealing and LD distributions are the same. In fact, annealing simply by reducing the temperature without changing the distribution at all can work, as long as it allows the LD to mix. The main insight we borrow from Song’s predictor-corrector method is that the reverse diffusion process offers a natural and effective annealing schedule which can enable successful mixing (even if the annealed distributions do not exactly match the distribution LD attempts to sample from). We show that in the SDE limit as $dt \to 0$, an (infinitesimal) step of the CFG-DDPM SDE is equivalent to an step of the DDIM SDE (the predictor) plus a step of the LD SDE on $p_{t, \gamma}$ (the corrector).
>
> **The interpretations of Theorem 1 and 2 are not clear stated. Is CFG-DDIM always tends to be sharper than CFG-DDPM, or it just because the special construction used in Theorem 1 and 2?**
>
> * Thank you, a similar point was raised by reviewer CU4e [copying the common response here]: Apologies, we did not intend to claim that DDIM-CFG is always exponentially sharper than DDPM; only that this is so for our particular counterexample/construction (hence our theorem statement that “there exists...”). We have made edits to the wording of Theorems 1 & 2, and also added a formal Theorem 4 to clarify any imprecision.
>
> **What is the potential usefulness of the derived results in further theoretical analysis of diffusion model?**
>
> * In terms of enabling future theoretical analysis, we believe that the generalized predictor-corrector framework (which can also be understood as a particular kind of annealed Langevin dynamics) is a useful perspective for analysis of a variety of settings that are related to, but not exactly, diffusion (and hence lack the theoretical guarantees of diffusion). In this framework the predictor is usually a reverse diffusion process, which is a natural and effective way to do the annealing, and the corrector can be flexibly chosen as “some distribution we hope to sample from/study”. For example, we discuss some specific ideas for alternative correctors in our response to reviewer qG7h’s question about “going beyond gamma-powered”. Please see also our response to reviewer qG7h’s question “What do we gain from writing out the SDE limit of PCG” for a general discussion of the usefulness of our theoretical analysis in understanding CFG specifically. Did this clarify your concern? If not, we are happy to discuss further.
>
> **Questions:**
>
> **Is it always true that a larger and more Langevin dynamic steps in the corrector can lead to sharper distribution?**
>
> * Re: Larger LD steps: larger steps fail to satisfy our theory, so we are not sure what happens in a formal mathematical sense. Moreover, large LD steps are likely to lead to instability in practice.
> * Re: More LD steps: In our experiments, more LD steps in the corrector typically increased sharpness. This can be seen in Figure 4, where increasing the number of Langevin steps appears to also increase the “effective” guidance strength. This is because the dynamics does not fully mix: one Langevin step ($K = 1$) does not suffice to fully converge the intermediate distributions to $p_{t,\gamma}$, but additional steps brings us closer to the fully-sharpened distribution. (Note also that CFG with guidance strength $\gamma$ corresponds to PCG with $K=1$ and guidance strength $2 \gamma - 1$. If we took many steps the PCG distribution would sharpen all the way to $2 \gamma - 1$, but with only a single step it makes only limited progress.)

---

> > ### Comment · Reviewer_q66d · 2024-11-27
> >
> > The response answers most of my question, I have increased my score to 6.

---

> > > ### Author Response · Authors · 2024-11-27
> > >
> > > Thank you!

---

### Official Review · Reviewer_aJqz · 2024-11-01

**Soundness:** 3
**Presentation:** 3
**Contribution:** 3
**Rating:** 6
**Confidence:** 3

**Summary:**

This paper provides a comprehensive, well-founded exploration of classifier-free guidance, establishing it as a viable, efficient alternative to classifier-based guidance methods. By grounding CFG in a predictor-corrector framework, the paper not only enhances understanding of diffusion models but also opens new paths for controlling generative models with minimal complexity.

**Strengths:**

### 1. Theoretical Foundation: By framing CFG within a mathematical context, the paper provides a rigorous basis for understanding its behavior and optimizing its use in diffusion models.

### 2. Experiment Validation: The paper provides the experiments to support its methodology.

**Weaknesses:**

### 1. The paper explains the classifier-free guidance, but I did not see whether your method can boost the performance of the diffusion model compared to DDPM, DDIM, or the consistency model.

### 2. I do not see the benefit of your understanding of CFG. Whether your understanding of CFG can benefit the theory results of CFG?

**Questions:**

### 1. Whether your method can boost the performance of the diffusion model compared to DDPM, DDIM, or the consistency model.

### 2. What's the benefit of your understanding of CFG? Whether your understanding of CFG can benefit the theory results of CFG in [Fu24]?

[Fu24] Unveil Conditional Diffusion Models with Classifier-free Guidance: A Sharp Statistical Theory

---

> ### Author Response · Authors · 2024-11-15
> **Responses to Weaknesses and Questions**
>
> We thank the reviewer for their careful review of our paper and helpful feedback. We address each point that was raised individually (original questions **bolded**). We also updated the PDF based on your feedback and that of other reviewers as detailed in the *Response to all reviewers*.
>
>
> **Weaknesses:**
>
> **1. The paper explains the classifier-free guidance, but I did not see whether your method can boost the performance of the diffusion model compared to DDPM, DDIM, or the consistency model.**
>
> * A similar point was raised by reviewer qG7h [copying the common response here]. As we discuss in Section 5.2, although we do present PCG primarily as a tool to understand CFG, the PCG framework outlines a broad family of guided samplers, which may be promising to explore in practice. For example, the predictor can be any diffusion denoiser, including CFG itself. The corrector can operate on any distribution with a known score, including compositional distributions, or any other distribution that might help sharpen or otherwise improve on the conditional distribution. Finally, the number of Langevin steps could be adapted to the timestep. Exploring this design space in order to improve on the practical performance of CFG is something we hope to explore in the future, that could help improve prompt-alignment, diversity, and quality.
>
> **2. I do not see the benefit of your understanding of CFG. Whether your understanding of CFG can benefit the theory results of CFG?**
>
> * Reviewer qG7h asked a similar question [copying the common response here]. CFG has been hugely impactful in practice but is not well grounded theoretically. It’s essentially a hack, and it’s not really clear why it should work at all. As our counterexamples demonstrate, the common intuition that CFG samples from $p_{0,\gamma}$ is not correct, and CFG does not even represent a valid reverse diffusion process. Our basic goal in this work is to explain why CFG is actually in some theoretical sense a “reasonable thing to do” and explain why we might expect it to work. We do so by showing an equivalence between CFG and a particular kind of annealed Langevin dynamics, where a conditional diffusion provides the annealing the schedule and the LD operates on $p_{t,\gamma}$.  This is a “reasonable” thing to do in the sense that if we ran LD to convergence (at least at the final step) we would be able to sample from the actual sharpened distribution $p_{0,\gamma}$, and even if we we only take one LD step (which is equivalent to CFG) we are at least making progress toward the sharpened distribution. Meanwhile, the diffusion provides an annealing schedule that enables the LD to mix. This connection casts CFG as at least a theoretically-grounded sampler (even though it’s not a true reverse diffusion sampler), and clarifies its relationship to sampling from $p_{0,\gamma}$ in terms of taking one LD step toward it within an annealing loop.
>
> **Questions:**
>
> **1. Whether your method can boost the performance of the diffusion model compared to DDPM, DDIM, or the consistency model.**
>
> * Please see response to first point in Weaknesses.
>
> **2. What's the benefit of your understanding of CFG? Whether your understanding of CFG can benefit the theory results of CFG in [Fu24]?**
>
> **[Fu24] Unveil Conditional Diffusion Models with Classifier-free Guidance: A Sharp Statistical Theory**
>
> * Note that [Fu24] uses the term CFG in a difference sense that we do here. By CFG we mean a sampler that uses a modified score, while [Fu24] uses CFG to refer to a particular training method that simultaneously parametrizes the unconditional and conditional scores. So we do not see an an immediate connection or application of our theory to the results of [Fu24].
> * That said, we wonder if Figure 6 in our work might be of interest to Fu et al., given their interest in how accurately a score function can be learned. In the example shown in Figure 6 we explore the impact of imperfectly-learned scores on generalization. We hypothesize that using CFG (in the sampling sense) to “sharpen” the sampled distribution could in some case improve generalization when the scores were imperfectly learned.  In our example, we consider a GMM with a dominant cluster. If we undersample this distribution during training, the learned model learns the dominant part of the distribution well, but it doesn't learn the non-dominant parts well, leading to poor samples in those regions. However, if we sample from the “sharpened” distribution with CFG (using those same imperfect scores), we do better, because the distribution we're trying sample from has most of its mass in regions that we did learn well.

---

> > ### Comment · Reviewer_aJqz · 2024-11-24
> >
> > Thanks for your response. The response answers my question. Thus, I keep my score as 6.

---

### Official Review · Reviewer_qG7H · 2024-11-04

**Soundness:** 3
**Presentation:** 3
**Contribution:** 3
**Rating:** 6
**Confidence:** 4

**Summary:**

This paper attempts to understand classifier-free guidance from a theoretical perspective. A special characteristic of classifier-free guidance is that it introduces a strength parameter $\gamma$ so that the plug-in score function is not precisely $\nabla \log p_t(x | y)$. Although practical results demonstrate promises of this methodology, its theoretical analysis is still largely missing. This paper presents new understanding of classifier-free guidance by first pointing out that the terminal distribution is hard to find. From my reading, this result is relatively a minor contribution. More interesting results come when connecting classifier-free guidance to predictor-corrector algorithm.

**Strengths:**

This paper is well written and the results appear to be correct and sound.

I am particularly appreciative of the discussions in Section 5, not only introduces relevant literature, but also touches on limitations and future directions.

Understanding classifier-free guidance from a theoretical perspective is an important direction.

**Weaknesses:**

Practical implication of the study may be limited.

**Questions:**

The following two recent works are related to guidance in diffusion models; they focus on mixture models. "What does guidance do? A fine-grained analysis in a simple setting" and "Theoretical Insights for Diffusion Guidance: A Case Study for Gaussian Mixture Models".

Algorithm 1 states that Line 4 is a DDIM step. From my understanding, DDIM (as well as DDPM) uses an exponential integrator to discretize the backward ODE (SDE). Line 4 is a Euler discretization. Some discussion might be needed.

What do we gain from writing out the SDE limit of PCG?

Are there practical reasons to sample from the gamma-powered distribution? I believe the gamma-powered distribution comes from the classifier-free guidance. In practice, people only aim to promote label alignment and keep high sample fidelity. Is it possible to go beyond the gamma-powered distribution?

---

> ### Author Response · Authors · 2024-11-15
> **Responses to Weaknesses**
>
> We thank the reviewer for their careful review of our paper and helpful feedback. We address each point that was raised individually (original questions **bold**). We also updated the PDF based on your feedback and that of other reviewers as detailed in the *Response to all reviewers*.
>
> **Weaknesses**:
> **Practical implication of the study may be limited.**
> * As we discuss in Section 5.2, although we do present PCG primarily as a tool to understand CFG, the PCG framework outlines a broad family of guided samplers, which may be promising to explore in practice. For example, the predictor can be any diffusion denoiser, including CFG itself. The corrector can operate on any distribution with a known score, including compositional distributions, or any other distribution that might help sharpen or otherwise improve on the conditional distribution. Finally, the number of Langevin steps could be adapted to the timestep. Exploring this design space in order to improve on the practical performance of CFG is something we hope to explore in the future, that could help improve prompt-alignment, diversity, and quality.

---

> ### Author Response · Authors · 2024-11-15
> **Responses to Questions**
>
> **Questions:**
>
> **The following two recent works are related to guidance in diffusion models; they focus on mixture models. "What does guidance do? A fine-grained analysis in a simple setting" and "Theoretical Insights for Diffusion Guidance: A Case Study for Gaussian Mixture Models".**
>
> * Thanks for pointing these out! They are indeed relevant to our Gaussian counterexamples (in fact the first one gives a theoretical analysis confirming our qualitative observations in the 2-cluster GMM counterexample, and actually cites us as well). We cited both in the new draft (end of section 3.2)
>
> **Algorithm 1 states that Line 4 is a DDIM step. From my understanding, DDIM (as well as DDPM) uses an exponential integrator to discretize the backward ODE (SDE). Line 4 is a Euler discretization. Some discussion might be needed.**
>
> * Great question and you’re absolutely right. Algorithm 1 uses a first-order Euler discretization, which is convenient for our mathematical analysis. Algorithm 2 (our suggestion for an explicit, practical implementation of PCG) uses the original DDIM discretization involving an exponential integrator, which is more common in practice, as you mentioned. (We discuss this in Appendix E in the updated PDF and it was present in footnote 2 of original PDF.)
>
> **What do we gain from writing out the SDE limit of PCG?**
>
> * CFG has been hugely impactful in practice but is not well grounded theoretically. It’s essentially a hack, and it’s not really clear why it should work at all. As our counterexamples demonstrate, the common intuition that CFG samples from $p_{0,\gamma}$ is not correct, and CFG does not even represent a valid reverse diffusion process. Our basic goal in this work is to explain why CFG is actually in some theoretical sense a “reasonable thing to do” and explain why we might expect it to work. We do so by showing an equivalence between CFG and a particular kind of annealed Langevin dynamics, where a conditional diffusion provides the annealing the schedule and the LD operates on $p_{t,\gamma}$.  This is a “reasonable” thing to do in the sense that if we ran LD to convergence (at least at the final step) we would be able to sample from the actual sharpened distribution $p_{0,\gamma}$, and even if we we only take one LD step (which is equivalent to CFG) we are at least making progress toward the sharpened distribution. Meanwhile, the diffusion provides an annealing schedule that enables the LD to mix. This connection casts CFG as at least a theoretically-grounded sampler (even though it’s not a true reverse diffusion sampler), and clarifies its relationship to sampling from $p_{0,\gamma}$ in terms of taking one LD step toward it within an annealing loop.
>
> **Are there practical reasons to sample from the gamma-powered distribution? I believe the gamma-powered distribution comes from the classifier-free guidance. In practice, people only aim to promote label alignment and keep high sample fidelity. Is it possible to go beyond the gamma-powered distribution?**
>
> * Yes, the gamma-powered distribution comes from CFG (and originally from classifier-guidance), and although it empirically works well, there could certainly be a distribution that works even better! As we discuss in section 5.2, one advantage of the PCG framework is that it outlines a broad family of guided samplers. In particular, the corrector could operate on any distribution with a known score, including compositional distributions, or any other distribution that might help sharpen or otherwise improve on the conditional distribution. We have done some limited exploration in this direction: we tried using distributions of the form $p(x|c)^\gamma)$, but it did not work well in our (limited) experiments. (We have some ideas about why, but they are out of scope for this particular paper) However, trying to find better sharpening distributions to promote label alignment and sample quality is a very interesting open direction; if we can find them, the PCG framework would enable us to exploit them.

---

> > ### Comment · Reviewer_qG7H · 2024-12-02
> >
> > Sorry for a delay in reply. I appreciate authors' responses to my questions and comments. I am happy to maintain my initial positive rating.

---

### Public Comment · ~Zhengqi_Gao1 · 2024-11-12
**A question about VE and VP**

Hi, I come across this paper today, and it is really inspiring and has huge contribution to the understanding of CFG. I really like the constructed counter examples. I have a quick/minor question which would like to seek the authors' feedback. In the main text, e.g., above Eq (3), the authors mentioned **they will mainly focus on VP setting. I read A.1 and it seems the proofs are done under the VE setting.** I wonder if the statement that the target distribution is not the titled one when using CFG still holds under the VP setting? Can we have a formal proof in those counter examples?

After reading the equation, I am also confused about the derivations. I hope the authors can correct me if I am wrong. Specifically, according to the forward and reverse denoising expression of VE:

Forward: $dx_t=\sqrt{\frac{d\sigma_t^2}{dt}}dw$, and Reverse: $dx_t=-\frac{d\sigma_t^2}{dt}\nabla logp_t(x_t)dt$ from https://arxiv.org/pdf/2405.21059.

The case shown in Appendix A.1 should correspond to $\sigma_t^2=0.5t^2$, so that the forward can reduce to $dx_t=\sqrt{t}dw$ as stated in A.1. When substituting $\sigma_t^2=0.5t^2$ into the reverse formula, we should have: $dx_t=-t\nabla logp_t(x_tdt)$, but the authors wrote $dx_t=-0.5\nabla logp_t(x_tdt)$ in Appendix A.1. Namely, **the term $t$ is missing.** Could the authors help me understand where I did wrong?

---

> ### Author Response · Authors · 2024-11-14
>
> Hey, thanks for the question and sorry for the confusion. First, yes you’re right, we use VP for the PCG algorithm but the counterexamples use a VE schedule (we'll edit the next draft to clarify this). And secondly there is a typo (which I just fixed): the VE SDE should simply read $dx = dw$! (which just corresponds to $\sigma_t^2 = t$). Sorry about that and thanks for noticing!

---

> > ### Public Comment · ~Zhengqi_Gao1 · 2024-11-20
> > **Thanks!**
> >
> > Thanks! This totally clarify my confusions. Good luck in the rebuttal!

---

### Author Response · Authors · 2024-11-15
**Response to all reviewers**

We thank all reviewers for their time. We have updated the PDF; the main changes are summarized here:

* We added a fully formal statement and proof of our claims, as Theorem 4 in the Appendix. We also edited the wording around Theorems 1 and 2 in the body.
* We added three remarks in Section 4.1, discussing the relationship between PCG and annealed Langevin dynamics more explicitly.
* We fixed typos in Algorithms 1 & 2 which cause misunderstanding.
* We added several exploratory experiments considering the effect of different discretization choices (Appendix D, Table 4).
* We added discussion of some related works pointed out by reviewers [Chidambaram et al. 24], [Wu et al. 24].

We will respond to each reviewer individually in the thread.

---

### Author Response · Authors · 2024-11-23
**Following-up on Rebuttals**

Dear reviewers,

Since we are nearing the end of the discussion period, we wanted to ask if your concerns have been adequately addressed by our rebuttals (& updated PDF). If there are any remaining concerns, we are happy to follow-up. Thank you again for your engagement throughout this process.

--Authors

---

### Meta-Review · Area_Chair_wb1k · 2024-12-20

**Metareview:**

The manuscript studies classifier-free guidance for conditional generation. From a theoretical point of view, the manuscript aims to clarifies some misconception of the classifier-free guidance model in the literature, and also provides some new perspective from the connection with prediction-correction schemes. As pointed out during the discussion phase, the misconception is well understood by experts in the field and there have been previous works on this point. While the connection with prediction-correction scheme is interesting, it is unclear to the meta-reviewer whether it leads to practical better schemes. Based on these, the meta-reviewer feels that the manuscript falls short of the bar of acceptance, after carefully reading the manuscript and the discussions.

**Additional Comments On Reviewer Discussion:**

The discussion is thorough and the authors answered most of the reviewers' questions during the discussion phase.

---

### Decision · Program_Chairs · 2025-01-22

Reject